# SynLaD: Latent Diffusion for Generating Synthesizable Molecules Conditioned on 3D Pharmacophore Profiles

**Miruna Cretu** [1] [2] [*]   **John Bradshaw** [2]   **Patricia Suriana** [2]   **Saeed Saremi** [2]   **Omar Mahmood** [2]   **Kirill Shmilovich** [2]
**Kangway Chuang** [2]   **Vishnu Sresht** [2]   **Colin Grambow** [2]

## Abstract

We present SynLaD, a latent diffusion framework for small-molecule generation that unifies ligand-based drug design objectives (what to make) with synthetic accessibility (how to make it). Current models typically optimize one objective at the expense of the other, creating a bottleneck for discovering high-scoring and synthesizable molecules. SynLaD combines reaction-constrained generation with pharmacophore-conditioned 3D design by learning a latent space that decodes to both 3D structures and synthesis pathways. An encoder maps molecules to a latent representation used by two decoder heads: (i) a geometric head that reconstructs atom types and coordinates and (ii) an autoregressive synthesis head that outputs synthetic routes in a serialized, reaction-based notation. A diffusion transformer generates novel latents in the learned space, conditioned on pharmacophore profiles. Across analogue generation tasks for bioactive ligands, SynLaD outperforms existing baselines in synthesizable and diverse *hit* generation, demonstrating that a single model can produce shape-aligned molecules with feasible synthesis plans.

## 1. Introduction

Ligand-based drug design (LBDD) uses known bioactive compounds to guide the design of new molecules with similar three-dimensional (3D) shapes and physicochemical properties (Acharya et al., 2011). In contrast to structure-based drug design (SBDD), which relies on accurate protein

structures and is often more computationally demanding, LBDD remains effective when target structures are unavailable and is therefore widely used for hit discovery and hit diversification (Grebner et al., 2020). Central to LBDD is the pharmacophore, which specifies the spatial arrangement of interaction features—such as hydrogen bond donors or acceptors, charged groups, and aromatic centers—responsible for protein-ligand complementarity.

A common LBDD strategy extracts pharmacophores from known ligands and uses them in virtual screening to identify molecules with similar shapes and features (Goodnow & Gillespie, 2007). Methods such as ROCS follow this approach and have shown competitive or better performance than structure-based methods (OpenEye Scientific Software, 2025; Hawkins et al., 2007; Sheridan et al., 2008). However, the rapid expansion of accessible chemical space (Kuan et al., 2023) renders exhaustive screening increasingly computationally prohibitive. Generative modeling has emerged as a promising alternative to brute-force search, allowing for faster inference times and the proposing of molecules outside of predefined spaces. Conditional generative models for 3D small molecule design span a range of families, from autoregressive (Peng et al., 2022) to diffusion (Luo et al., 2021; Schneuing et al., 2024; Guan et al., 2023; Cremer et al., 2024), flow matching (Schneuing et al., 2025; Cremer et al., 2025), and variational autoencoders (Zhu et al., 2023). While performing well at capturing distributions of desired molecular motifs, these models often produce synthetically inaccessible molecules (Gao & Coley, 2020; Stanley & Segler, 2023).

Synthesis-aware generative methods for designing new molecules have typically either treated synthesizability as an additional property to optimize (Liu et al., 2022; Guo & Schwaller, 2025; Gómez-Bombarelli et al., 2018) or enforced it by generating molecules via synthetic pathways/reactions rather than as unconstrained molecular graphs (Hartenfeller et al., 2012; Vinkers et al., 2003; Horwood & Noutahi, 2020; Bradshaw et al., 2019b; Gao et al., 2022; Swanson et al., 2024; Cretu et al., 2025). However, these approaches are not conditioned on three-dimensional chemical features, which are critical for protein-ligand bind-

---

[*]Work done during an internship at Prescient Design.
[1]University of Cambridge, Cambridge, UK [2]Prescient Design (AI for Drug Discovery), Genentech, South San Francisco, USA. Correspondence to: Miruna Cretu <mtc49@cam.ac.uk>, John Bradshaw <bradshaw.john@gene.com>, Colin Grambow <grambow.colin@gene.com>.

*Proceedings of the $43^{rd}$ International Conference on Machine Learning*, Seoul, South Korea. PMLR 306, 2026. Copyright 2026 by the author(s).

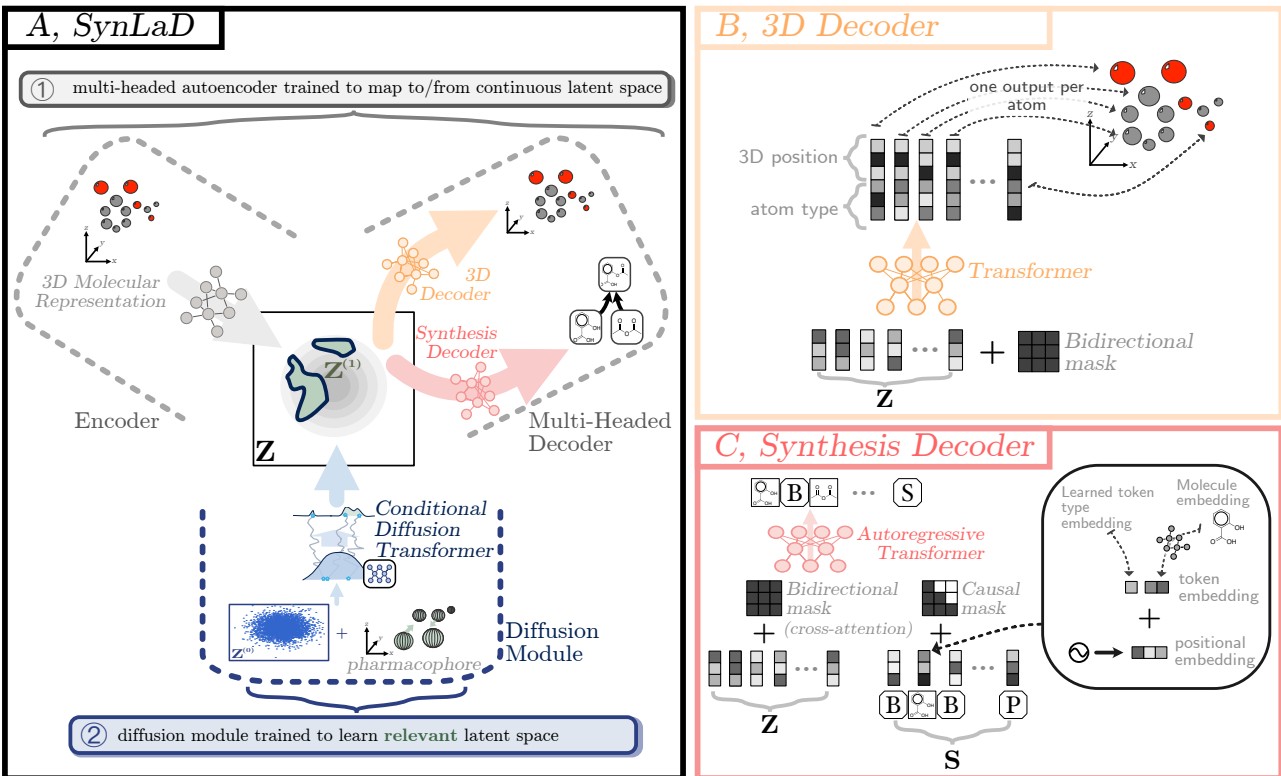

*Figure 1.* **A, Overview of SynLaD: a multi-head generative model that jointly produces 3D molecular structures and synthesis plans.** SynLaD is trained in two stages: ① an autoencoder learns a latent representation of 3D molecules, with an auxiliary decoder that generates a synthesis plan for each structure. ② a pharmacophore-conditioned diffusion model learns to sample latents that decode to molecules containing a desired pharmacophore. **B, 3D Decoder:** a transformer with a bidirectional mask maps latent representations to atom types and 3D coordinates. **C, Synthesis Decoder:** an autoregressive transformer maps latent representations to serialized synthesis plans. At inference time, the synthesis decoder is conditioned on the 3D-informed latent via cross-attention, enabling SynLaD to generate outputs that inherit pharmacophore-aware structural information while remaining directly tied to a synthesizable route.

ing. As a result, bioactive ligand generation is typically framed either as a reward-based problem using binding-affinity proxies (Horwood & Noutahi, 2020; Cretu et al., 2025) or as conditioning on known ligands via chemical space projection (Luo et al., 2024).

We bridge the gap between 3D chemical feature conditioning and synthesizable molecule generation with SynLaD (*Synthesis-aware Latent Diffusion*), a conditional latent diffusion model that decodes latents to molecules in both 3D and synthesizable space. An overview of SynLaD is shown in Figure 1. At its core, SynLaD employs a molecular variational autoencoder with a dual-decoder architecture: a single encoder maps a 3D molecule to the latent space, from which one decoder reconstructs the 3D atomic coordinates and corresponding atom types, while a second autoregressive decoder generates an associated synthesis pathway. Joint training of these decoders couples spatial fidelity and synthesizability in the learned latent representation.

Conditional generation is then performed in the learned latent space using a Diffusion Transformer (DiT) (Rombach et al., 2022; Joshi et al., 2025). The compressed, seman-

tically rich latent space enables efficient sampling and improved conditional control (Rombach et al., 2022; Liu et al., 2023), while also allowing SynLaD to generate structure-informed embeddings that the synthesis head decodes into synthesis pathways. This introduces variability (i.e., different conditioning inputs to the synthesis decoder), which we demonstrate is essential for ensuring sample diversity. Contrary to recent synthesis-constrained generative models (Swanson et al., 2024; Cretu et al., 2025; Koziarski et al., 2024; Gao et al., 2025; Lee et al., 2026) which rely on reaction templates, SynLaD uses a reaction prediction model trained on diverse, complementary chemical transformations to infer products. By removing the reliance on fixed templates, our approach mitigates the template bottleneck that can restrict designs to a narrow, predefined chemical space, and supports broader exploration via more general molecular-graph edits, and the possibility of extrapolation.

We evaluate SynLaD using standard molecule quality metrics, including pose validity and synthesizability, under both unconditional and pharmacophore-conditioned generation. We further assess its practical utility on bioactive analogue

generation for ligands from the Lit-PCBA benchmark (Tran-Nguyen et al., 2020). Across settings, SynLaD matches the molecule-quality profile of its separately trained counterparts while consistently improving the fraction of samples with plausible synthetic routes, and it outperforms existing baselines in producing synthesizable, diverse hits under pharmacophore-guided sampling.

The contributions of this work are thus: (1) a dual-constrained latent space that couples two complementary but often competing representations—3D molecular structure and synthesis plans—encouraging latents to remain simultaneously decodable into valid geometries and executable routes; (2) a pharmacophore-conditional latent generative model that enables targeted, 3D-aware design; and (3) an end-to-end evaluation protocol that jointly reports diversity, synthesizability, and target-conditioned hit discovery trade-offs—explicitly contrasting our conditional generation method with library screening and other amortized and non-amortized generative baselines. We release the code for SynLaD at https://github.com/prescient-design/synlad.

**Conflict of Interest Disclosure.** The authors JB, PS, SS, OM, KS, KC, VS, and CG are employed by Genentech, Inc., a member of the Roche Group, and developed SynLaD, the model proposed and evaluated in this paper. MC was an intern at Genentech, Inc. during the development of this work.

## 2. Background and related work

**Structure- and ligand-based drug design.** Many generative models have been proposed to sample small molecules in sequence (Gupta et al., 2018; Blaschke et al., 2020; Segler et al., 2018), 2D (Jensen, 2019; Jin et al., 2017; QIN et al., 2025), 3D (Le et al., 2024; Dunn & Koes, 2024; Huang et al., 2025; Irwin et al., 2025; Vonessen et al., 2025), and voxel spaces (O. Pinheiro et al., 2023). Conditional generation for protein binders can be broadly categorized into pocket-conditioned approaches, which condition on binding-site representations and implicitly capture protein-ligand interactions, and interaction-conditioned approaches, which condition directly on desired interaction cues such as 3D shape, pharmacophores, or electrostatic potential surfaces. The latter is most relevant to our work. Several methods generate SMILES strings or molecular graphs conditioned on 3D pharmacophore representations (Imrie et al., 2021; Zhu et al., 2023; Xie et al., 2025; Mahmood et al., 2025). MolSnapper (Ziv et al., 2025) and, more recently, ShEPhERD (Adams et al., 2025) instead condition on richer interaction profiles and generate molecules directly in 3D space. However, MolSnapper supports a limited set of pharmacophore types, and neither method explicitly constrains generation to synthesizable chemical space.

**Synthesis-aware generation.** Parallel work has focused on enhancing synthesizability by generating molecules via reaction pathways. Composing these reaction pathways into full synthesis plans is a challenging discrete search problem, and existing methods span diverse paradigms, including reinforcement learning (Gottipati et al., 2020; Horwood & Noutahi, 2020), search (Swanson et al., 2025), surrogate-guided optimization (Korovina et al., 2020), genetic algorithms (Lo et al., 2025), and GFlowNets (Cretu et al., 2025; Koziarski et al., 2024; Seo et al., 2025). One particularly effective strategy is to cast synthesis plans as sequences and learn *serialized* pathway representations autoregressively (Bradshaw et al., 2020; Gao et al., 2022; Lee et al., 2026; Luo et al., 2024; Gao et al., 2025), analogous to next-token prediction in language modeling (Vaswani et al., 2017; Bengio et al., 2003; Sutskever et al., 2014). By amortizing the cost of discrete search, such models learn mappings from continuous representations to synthesis plans and can be integrated into larger frameworks (e.g., encoder-decoder architectures) for tasks such as molecular optimization or retrosynthesis (i.e., predicting a route for a given target molecule). While these approaches enforce viable synthesis routes, they do not directly control final 3D geometry. Recently, CGFlow (Shen et al., 2025) and SynCoGen (Rekesh et al., 2025) jointly design synthesis pathways and 3D poses, but they rely on fixed reaction template libraries and work in ambient space.

**Latent diffusion models.** Latent diffusion models (Vahdat et al., 2021; Rombach et al., 2022) perform diffusion (Sohl-Dickstein et al., 2015; Song & Ermon, 2019) in the learned latent space of an autoencoder rather than directly in the high-dimensional input space, enabling more efficient training and sampling. This paradigm has been highly effective in image, audio, and video generation, especially when combined with Diffusion Transformers (DiTs) (Peebles & Xie, 2023), which show that standard transformer backbones scale effectively as denoisers. In molecular modeling, Xu et al. (2023) introduced latent diffusion in the space of an equivariant autoencoder, while Joshi et al. (2025) proposed a unified (non-equivariant) latent diffusion model for small molecules and periodic materials. However, these approaches do not demonstrate latent generation conditioned on rich, chemically grounded features. We address this gap by enabling pharmacophore-conditioned latent generation and augmenting our autoencoder with an auxiliary synthesis-decoder head that produces explicit reaction pathways, jointly targeting native-3D conditional generation and synthesizability.

## 3. Methods

We present SYNLAD—a latent diffusion (Rombach et al., 2022) model for *de novo* small molecule generation conditioned on pharmacophore features (see Figure 1). SYNLAD

is trained in two stages: In stage 1, a variational autoencoder (Kingma & Welling, 2013; Rezende & Mohamed, 2015) encodes molecules into a shared latent space and a two-headed decoder reconstructs both the 3D molecular structure and a synthesis pathway. In stage 2, we train a Diffusion Transformer (Peebles & Xie, 2023) to generate new samples from the latent space, which are decoded into both 3D molecules and synthesis pathways. The two decoders are trained to reconstruct the *same* molecule under two *different* representations and at inference time the synthesis decoder is used to ensure the synthesizability of generated designs.

We hypothesize that this two-stage learning process offers two advantages. First, the shared latent space aligns the two modalities—molecular conformations and synthesis pathways—by requiring both decoders to reconstruct the same molecule from a single latent representation. Second, as shown in Joshi et al. (2025), operating in a latent space reduces the complexity of generation and enables explicit pharmacophore conditioning by decoupling discrete and continuous spaces.

### 3.1. Multi-headed autoencoder for multi-modal reconstruction

Our autoencoder consists of one encoder and two decoders trained using a combined reconstruction loss with a regularization term. We base our implementation of the 3D encoder and decoder on Joshi et al. (2025) and represent molecules using discrete atom types $\boldsymbol{A} = \{a_i\}_{i=1}^N \in \mathbb{Z}^{1 \times N}$ and continuous coordinates $\boldsymbol{X} = \{x_i\}_{i=1}^N \in \mathbb{R}^{3 \times N}$, where $N$ is the number of atoms.

#### 3.1.1. 3D ENCODER AND DECODER

Given a molecule represented as above, the encoder $f$ projects $\boldsymbol{A}$ and $\boldsymbol{X}$ into a per-atom latent representation $\boldsymbol{Z} = f(\boldsymbol{A}, \boldsymbol{X})$, and the 3D decoder $g_{\text{3D}}$ reconstructs the molecule from the latent, giving $\boldsymbol{A}', \boldsymbol{X}' = g_{\text{3D}}(\boldsymbol{Z})$, where $\boldsymbol{Z} = \{z_i\}_{i=1}^N \in \mathbb{R}^{d \times N}$. We use a standard transformer (Vaswani et al., 2017) and learn molecular symmetries via random rotations; further details on how atoms and dimensions are sampled at training/inference time is provided in Appendix C.1.

#### 3.1.2. SYNTHESIS DECODER

We define an auxiliary decoder $g_{\text{syn}}(\boldsymbol{Z})$, which, given a latent $\boldsymbol{Z}$, generates a synthesis pathway for the molecule represented in 3D by $(\boldsymbol{A}, \boldsymbol{X}) = g_{\text{3D}}(\boldsymbol{Z})$. A synthesis pathway can be represented using a directed acyclic graph (DAG), where nodes represent molecules and edges represent reactions (see Figure 1). Such a DAG shows how parent building block molecules (source nodes) react to first form more complex intermediate products and eventually a final product (a single sink node). To model such a DAG we

can *serialize* it, in a bottom-up manner similar to Bradshaw et al. (2020), into a linear sequence of tokens following the recipe in Appendix B.2. Once we have such a synthesis sequence $\mathbf{S} = (s_1, \dots, s_L)$, we train a causal transformer model to generate it by autoregressively modeling the conditional probability $p(s_i | \boldsymbol{S}_{<i}, \boldsymbol{Z})$. We embed each token in the sequence using a custom embedding scheme (see Appendix B.3) and train the transformer using teacher forcing conditioned with cross-attention on the latents.

**Product identity.** In our serialized synthesis plan representation, the identity of each product node is determined by the identities of the corresponding reactant molecules. During training, we use a dataset containing full synthesis plans (see Section 4), allowing the correct reaction products to be inserted directly. At test time, however, product information is not available and must be inferred. We therefore employ a reaction-prediction oracle that can predict the most likely product given a set of reactants. The oracle is a transformer-based encoder-decoder model built off the BART architecture (Lewis et al., 2020), closely related to the Molecular Transformer of Schwaller et al. (2019). Details of its training are provided in Appendix B.7.

#### 3.1.3. TRAINING OBJECTIVE

The two decoders are jointly trained with a weighted loss: the 3D decoder uses cross-entropy and MSE losses to reconstruct atom types and coordinates, while the synthesis decoder is optimized with a cross-entropy loss between its output logits and the ground-truth token sequence:

$$\mathcal{L} = \lambda \mathcal{L}_{\text{3D}} + (1 - \lambda)\mathcal{L}_{\text{synthesis}}$$
$$+ \beta D_{\text{KL}}\Big(\mathcal{N}(\mu_{\mathbf{Z}}, \sigma_{\mathbf{Z}}) \,\Big\|\, \mathcal{N}(0, I_d)\Big) \quad (1)$$

where:

$$\mathcal{L}_{\text{3D}} = \frac{1}{N}\sum_{i=1}^N H(a_i, \hat{a}_i) + \frac{1}{3N}\sum_{i=1}^N \frac{\|\tilde{x}_i - \hat{\tilde{x}}_i\|^2}{\sigma^2}, \quad (2)$$

$$\mathcal{L}_{\text{synthesis}} = \frac{1}{L}\sum_{i=1}^L H(s_i, \hat{s}_i). \quad (3)$$

Here, $\tilde{x}$ represents zero-centered coordinates, $\sigma^2$ is 1.0, $0 \leq \lambda \leq 1$ is the linear interpolation parameter between $\mathcal{L}_{\text{synthesis}}$ and $\mathcal{L}_{\text{3D}}$, $L$ is the total sequence length, and $s_i$ is the ground-truth synthesis token at step $i$. Following Higgins et al. (2017), Rombach et al. (2022), and Joshi et al. (2025) we introduce a per-channel KL-penalty weighted by $\beta$ that regularizes the learned latent toward a standard normal distribution.

### 3.2. Conditional generative modeling of latent representations

We train a flow matching model (Lipman et al., 2023; Song & Ermon, 2019; Ho et al., 2020; Joshi et al., 2025) on the latent space learned by our autoencoder, by denoising

latent samples from a base Gaussian distribution towards a target distribution represented by latents of ground-truth molecules.

**Pharmacophore embeddings.** A pharmacophore abstracts components of a molecule into pre-defined key interaction features, providing a compact representation of interaction-relevant moieties and their spatial coordinates. We represent pharmacophores using six discrete pharmacophore types $\boldsymbol{P} = \{p_i\}_{i=1}^{N_p} \in \mathbb{Z}^{1 \times N_p}$ (hydrogen bond donor, hydrogen bond acceptor, cation, anion, aromatic ring and hydrophobe) and coordinates for each pharmacophore feature present in the molecule: $\boldsymbol{X}^{\text{ph}} = \{x_i^{\text{ph}}\}_{i=1}^{N_p} \in \mathbb{R}^{3 \times N_p}$. These are extracted from the molecule conformation through a deterministic mapping of the pharmacophore's atom positions. The features are embedded into a unified learned representation using a standard transformer (see Appendix B.6), and this embedding serves as conditioning for the diffusion model via cross-attention. The parameters of the pharmacophore transformer encoder and those of the DiT are jointly optimized via the conditional flow matching loss:

$$\mathcal{L}_{\text{CFM}} = \mathbb{E}_{\substack{(\boldsymbol{P}, \boldsymbol{X}^{\text{ph}}, \boldsymbol{Z}_1) \sim p_{\text{data}}, \\ t \sim \mathcal{U}(0,1),\ \boldsymbol{Z}_0 \sim \mathcal{N}(0,I)}} \Big[ \|u_\theta(\boldsymbol{Z}_t,\ t,\ \tau_\theta(\boldsymbol{P}, \boldsymbol{X}^{\text{ph}})) \\ - v_t(\boldsymbol{Z}_t \mid \boldsymbol{Z}_0, \boldsymbol{Z}_1)\|_2^2 \Big], \quad (4)$$

where $u_\theta$ is the denoiser backbone, which predicts the vector field based on the current state $z_t$, the time step $t$, and the pharmacophore conditional embedding $\tau_\theta(\boldsymbol{P}, \boldsymbol{X}^{\text{ph}})$. $v_t$ is the ground-truth vector field and the state $\boldsymbol{Z_t} = (1-t)\boldsymbol{Z_0} + t\boldsymbol{Z_1}$ is a linear interpolation between a clean latent sample $\boldsymbol{Z_1}$ and a noise sample $\boldsymbol{Z_0}$ drawn from a standard normal distribution $\mathcal{N}(0, I)$. The time step $t$ is sampled uniformly from $\mathcal{U}(0, 1)$. Leveraging the equivalence between the velocity and endpoint flow matching formulations (Lipman et al., 2023), the model receives $\boldsymbol{Z_t}$ as input and predicts the terminal points of the trajectory during training.

Following Joshi et al. (2025), we use a Diffusion Transformer (DiT) (Peebles & Xie, 2023) as our denoiser architecture with self-conditioning (Yim et al., 2023) and adaptive layer norm for time-step $t$ modulation. We additionally introduce a cross-attention conditioning mechanism (trained with a conditioning dropout probability of 20%) to sample using classifier-free guidance during inference (Ho & Salimans, 2021):

$$\hat{u}_\theta(\boldsymbol{Z}_t, t, \tau_\theta(\boldsymbol{P}, \boldsymbol{X}^{\text{ph}})) = (1+w)u_\theta(\boldsymbol{Z}_t, t, \tau_\theta(\boldsymbol{P}, \boldsymbol{X}^{\text{ph}})) \\ - wu_\theta(\boldsymbol{Z_t}, t, \emptyset) \quad (5)$$

where $\hat{u}_\theta$ is the final vector field prediction. This is a linear combination of the conditional and unconditional predictions, where $\emptyset$ represents the null token used for the dropped-out condition. The guidance scale $w$ controls the strength of the conditioning signal and we experiment with both $w = 0$ and $w > 0$.

## 4. Experiments

**Dataset.** We train SynLaD on a set of reaction pathways extracted from the USPTO dataset (Lowe, 2017). We detail the construction of the dataset in Appendix B.1. Our dataset contains 67,512 synthesis pathways, each with 1 to 6 intermediary reactions. For each final product of the pathways, we compute 10 low-energy conformers using OpenEye's conformer generation software Omega (Hawkins et al., 2010; Friedrich et al., 2017). Our dataset contains synthesis products with 10 to 40 heavy atoms.

**Metrics.** We evaluate SynLaD on its ability to generate valid, novel, and realistic molecules. Sampled molecules generated by SynLaD are assessed using validity, internal diversity (Polykovskiy et al., 2020), and Fréchet ChemNet Distance (FCD) (Preuer et al., 2018) (definitions provided in Appendix B.8). Diversity is further measured by the number of unique Bemis-Murcko scaffolds (Bemis & Murcko, 1996), defined as the molecule's ring systems and connecting linkers with all side chains removed. Synthesizability is additionally evaluated using AiZynthFinder (Genheden et al., 2020), and pose quality is assessed with the PoseBusters benchmark (Buttenschoen et al., 2024) for molecules generated by our 3D decoder. For pharmacophore-conditioned experiments, we use OpenEye's ROCS scoring functions (Hawkins et al., 2007) to evaluate shape and pharmacophore overlap via Tanimoto Shape and Tanimoto Color scores, each ranging from 0 to 1 (with 0 representing no overlap and 1 representing perfect overlap), whose sum defines the Tanimoto Combo score. Following Grebner et al. (2020), we define a *hit* as a molecule with a Tanimoto Combo score of at least 1.2 relative to a query.

As described in Section 3.1.2, we use a reaction-prediction oracle to infer product information from sets of reactant tokens sampled by our decoder. The oracle is trained separately; implementation details and accuracy results are provided in Appendix B.7, where it achieves a top-1 product prediction accuracy of 84.6%.

### 4.1. Joint conformer and synthesis latent space

We first investigate reconstruction performance of the joint autoencoder. We randomly select 1000 molecules from the test set, encode them into our latent space, and decode each latent using both decoders. For the 3D decoder, we report match rate and RMSD between ground-truth and predicted coordinates (see Appendix B.8); for the synthesis decoder, we report match rate and 2D fingerprint Tanimoto similarity between the predicted end product and the ground-truth synthesis outcome. We ablate the latent dimension during training and compare inference strategies for the synthesis decoder (see Appendix B.4). Results are summarized in Table 1 for both *sampling* and *beam search* decoding. *Sampling* generates $N$ independent candidate sequences by

temperature-scaled stochastic sampling at each step, optionally using *top-k* truncation by setting logits of tokens outside of the top $k$ to $-\infty$. In contrast, *beam search* deterministically expands the $b$ (number of beams) most probable partial sequences. In our experiments, we set $N = 1$ and $k = 10$ for sampling and compare against beam search with $b = 5$. Beam search consistently outperforms top-$k$ sampling and is therefore used for latent-space sampling in subsequent experiments.

*Table 1.* **Autoencoder reconstruction accuracies.** Results are reported for two synthesis-decoder inference strategies, with synthesis match rate and RDKit fingerprint Tanimoto similarity computed between the final products of predicted and ground-truth synthesis sequences.

| Inference method | 3D metrics | | Synthesis metrics | |
| --- | --- | --- | --- | --- |
| | RMSD (Å)↓ | Match Rate (%)↑ | Match Rate (%)↑ | Tanimoto similarity↑ |
| Sampling | 0.05 | 98.5 | 43.4 | 0.73 |
| Beam search | 0.05 | 98.5 | 63.4 | 0.84 |

## 4.2. Unconditional generation

We next train an unconditional latent DiT denoiser using latents produced by our joint VAE (variational autoencoder) described in Section 4.1; model parameters and training details are provided in Appendix B.5. We sample using $T = 100$ ODE integration steps and decode the resulting latent with both decoders. Results are reported in Table 2, where we compare against (1) a latent diffusion model trained without a synthesis decoder head (equivalent to Joshi et al. (2025)) and (2) an unconditional autoregressive synthesis-decoder-only head. The results show that jointly training the two decoders does not degrade synthesizability or PoseBusters performance, and yields good validity, diversity, and novelty overall. Interestingly, joint training improves the synthesizability of molecules decoded by the 3D decoder, suggesting that the synthesis decoder steers generation toward latent regions more likely to decode to synthesizable molecules. Both SynLaD decoders significantly outperform the synthesis-unconstrained baseline in synthesizability although trained on the same data.

**Consistency analysis.** To enable pharmacophore-conditioned sampling of synthesizable molecules that also match a desired 3D profile, we evaluate how well the two decoders remain aligned on *generated* samples. Starting from a latent $Z$ sampled from the trained diffusion model, we decode both a 3D molecule and a synthesis pathway and quantify how often these outputs correspond to the same, or closely matching, chemical features. We report cross-decoder agreement via exact identity and chemical/shape similarity in Table 2. The synthesis-decoded products preserve the 3D shape and pharmacophore characteristics implied by the 3D-decoded structures. We show examples of output pairs in Appendix 7.

## 4.3. Pharmacophore-conditioned molecule generation

**In-distribution pharmacophore conditioning.** We evaluate the conditional generation performance of SynLaD by selecting 50 random *query* molecules from the test set and extracting their pharmacophores. For each query, we generate 100 candidate molecules with SynLaD conditioned on the extracted pharmacophore, and, as a baseline, compare against 5,000 randomly selected molecules from the training set. All candidates—whether sampled from the training set or generated by SynLaD—are evaluated using conformer enumeration with Omega, followed by shape and pharmacophore overlay scoring against the query molecule (see Appendix B.9). For consistency, we apply the same conformer enumeration to 3D-decoded outputs, even though these samples already exhibit good conformer alignment to the query (see Appendix Figure 10). For each molecule, we retain the highest scoring conformer and assess whether it is a hit or not (i.e., with a Tanimoto combo score $\geq 1.2$ to the query). We report the median number of hits and unique scaffold hits in Figure 3 and distributions of Tanimoto shape and color scores, as well as synthesizability scores assessed using AiZynthFinder (further metrics in Appendix Table 12). Results show that SynLaD generates samples with significantly higher 3D pharmacophore and shape similarities to the query than the baseline, while the synthesis decoder yields molecules with high synthesizability.

**Screening case study.** To assess SynLaD in a practical drug-discovery setting, we compare it against brute-force ROCS screening over our dataset, which serves as a proxy for a molecular library and reflects standard but time-consuming ligand-based screening. We randomly select 10 query molecules with more than 15 heavy atoms from our test set and calculate ROCS scores against all other molecules in the dataset ($\sim$67k molecules), excluding the query molecules themselves. In parallel, we perform pharmacophore-conditional generation with SynLaD, generating 1000 samples per query. For this experiment, we consider only outputs of the synthesis decoder of SynLaD, ensuring that outputs are synthesizable and directly applicable for experimental testing. We report hit counts for

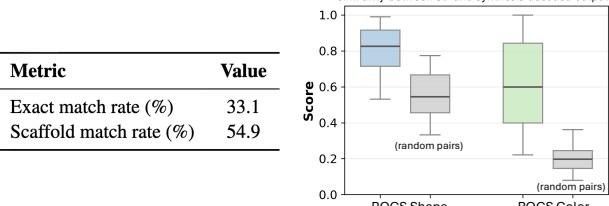

| Metric | Value |
| --- | --- |
| Exact match rate (%) | 33.1 |
| Scaffold match rate (%) | 54.9 |

*Figure 2.* **Cross-decoder agreement on generated samples.** We evaluate similarities between synthesis- and 3D-decoded molecules obtained from the same latent. As a control, we also report ROCS Tanimoto Shape/Color scores for randomly paired synthesis- and 3D-decoded molecules, illustrating the similarity expected in the absence of cross-decoder coupling.

*Table 2.* **Unconditional generation results.** ↑/↓ indicate that higher/lower is better. All metrics (see Appendix B.8) are evaluated over 1000 samples, except for AiZynthFinder (100 samples).

| | Method | Valid.↑ | IntDiv.↑ | Nov.↑ | FCD↓ | PB↑ | SA↓ | AiZynth.↑ |
|---|---|---|---|---|---|---|---|---|
| *separately-trained heads* | 3D Decoder | 88.5 | 86.0 | 100 | 2.91 | 0.89 | 2.78 | 0.59 |
| | Synthesis Decoder | 100 | 88.9 | 52.7 | 2.10 | - | 2.30 | 0.76 |
| *jointly-trained heads* | SynLaD (3D outputs) | 90.0 | 86.0 | 100 | 2.70 | 0.89 | 2.65 | 0.72 |
| | SynLaD (synthesis outputs) | 100 | 88.2 | 84.6 | 1.52 | - | 2.30 | 0.87 |

| Method | Hits | Unique scaff. hits | Max score | AiZynth. | Synthesizable hits |
|---|---|---|---|---|---|
| Dataset baseline | 0 | 0 | 1.14 | 0.90 | 0 |
| SynLaD (3D out) | 36.5 | 13.5 | 1.92 | 0.44 | 18.5 |
| SynLaD (syn out) | 29 | 9.5 | 1.88 | 0.80 | 25.0 |

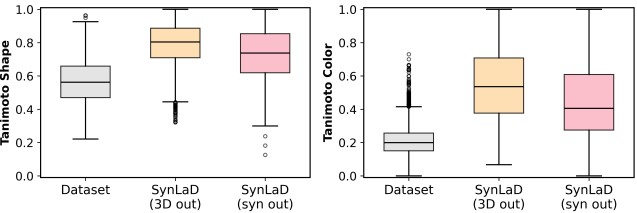

*Figure 3.* **Pharmacophore conditioned generation.** Metrics are evaluated for 50 random conditioning queries from a held-out set and 100 samples from SynLaD for each conditioning pharmacophore. *Left*: Values for hits, unique scaffold hits, and max score are medians across the 50 query molecules, while AiZynth is a mean (higher is better). Max score represents the maximum Tanimoto combo score for generated samples. *Right*: Distributions of ROCS shape and color (pharmacophore) similarity scores to query molecules (higher better).

both methods in Table 3, and per-query results in Appendix Figure 8. SynLaD achieves higher hit counts than the typical brute-force baseline in ligand-based screening, while using nearly two orders of magnitude fewer samples (1k molecules) than the size of the screened library (∼67k).

*Table 3.* **Comparison to library screen.** Number of hits averaged over 10 queries from the test set. Here, we compare against the synthesis decoder of SynLaD as hits are more likely to be synthesizable and therefore able to be experimentally tested.

| Method | Hits (avg.) |
|---|---|
| Dataset screen | 49.7 |
| SynLaD | 59.4 |

**Bioactive hit diversification.** We next investigate the out-of-distribution performance of SynLaD on a bioactive hit diversification task, which aims to find structural analogues of known bioactive compounds. This mirrors real-world drug discovery practices, where diverse candidates are needed to iteratively optimize activity while satisfying synthesizability, developability, and other constraints. We select ten targets from the Lit-PCBA benchmark (Tran-Nguyen et al., 2020) (see Appendix C.2) and compare against ShEPhERD (Adams et al., 2025), SynFormer (Gao et al., 2022), and REINVENT (Blaschke et al., 2020; Loeffler et al., 2024). ShEPhERD is an SE(3)-equivariant diffusion model that jointly denoises 3D molecular graphs and representations of their shapes and interaction profiles; we compare against their inpainting setting which conditions on electrostatic potential surfaces and pharmacophores. SynFormer is a transformer-based encoder-decoder framework that generates synthesizable 2D molecules in reaction-template and building-block space; we specifically compare against their SynFormerED variants for amortized analogue generation. As an orthogonal baseline, following Papadopoulos et al.

(2021), we fine-tune REINVENT (Blaschke et al., 2020) using a ROCS Tanimoto color score as reward function. Unlike SynLaD and the other baselines, REINVENT does not perform amortized sampling and instead requires reinforcement learning-based optimization per query, making the comparison methodologically orthogonal (similar techniques can also be applied to autoencoders (Tripp et al., 2020)). REINVENT is also substantially more computationally expensive, requiring approximately 6 hours per query to generate the optimized samples here, compared to ∼1 min for SynLaD, ≤1 min for SynFormer, and ∼28 min for ShEPhERD (Appendix C.4) per 100 samples, but we include it as it is a strong baseline in molecular generation.

For each query, we generate 500 molecules by conditioning on the pharmacophore profile of the native ligand in its bound conformation. We apply the same evaluation pipeline as in the previous section and report distributions of aggregated Tanimoto shape and color scores across all ten queries, along with averaged molecule validity, synthesizability, and hit counts, in Figure 4 and Table 4. Note that AiZynthFinder is reported for subsets (100 samples due to compute cost) of all generated molecules, not just hits. SynLaD achieves a strong overall balance, producing a high number of synthesizable hits, succeeding on a large fraction of queries, and maintaining high validity. Compared to the other synthesis-constrained baseline, SynFormer, SynLaD's synthesis decoder produces candidates with consistently higher ROCS shape and color overlap, suggesting that the explicit 3D representation our model learns improves conditional design; it also yields more hits and higher retrosynthesis success. The 3D outputs of SynLaD outperform ShEPhERD on most queries in terms of pharmacophore overlap, hit count, and synthesizability, while achieving comparable shape overlap. The synthesis-decoder output also outperforms ShEPhERD

*Table 4.* **Bioactive hit diversification.** We sample 500 samples per query for all methods. We report averaged metrics over all ten queries and AiZynthFinder success rate for a random subset of 100 generated molecules. Amortized methods do not require retraining for each target. Max score refers to the maximum Tanimoto combo score registered.

| Method | Validity | Hits (avg.) | Unique scaff. hits (avg.) | Max score | AiZynth. | Num. queries w/ $\geq$1 hit | Amortized | Synthesizable hits (avg.) |
|---|---|---|---|---|---|---|---|---|
| REINVENT | 0.98 | 21.1 | 3.7 | 1.15 | **0.75** | 3 | ✗ | **16.3** |
| ShEPhERD | 0.55 | 15.7 | 13.0 | 1.23 | 0.18 | 6 | ✓ | 6.3 |
| SynFormer | **1.0** | 13.8 | 6.7 | 1.31 | 0.50 | **8** | ✓ | 6.7 |
| SynLaD (3D out) | 0.49 | **38.1** | **18.8** | **1.45** | 0.22 | **8** | ✓ | 12.7 |
| SynLaD (syn out) | **1.0** | 17.9 | 6.9 | 1.30 | **0.75** | 5 | ✓ | 14.6 |

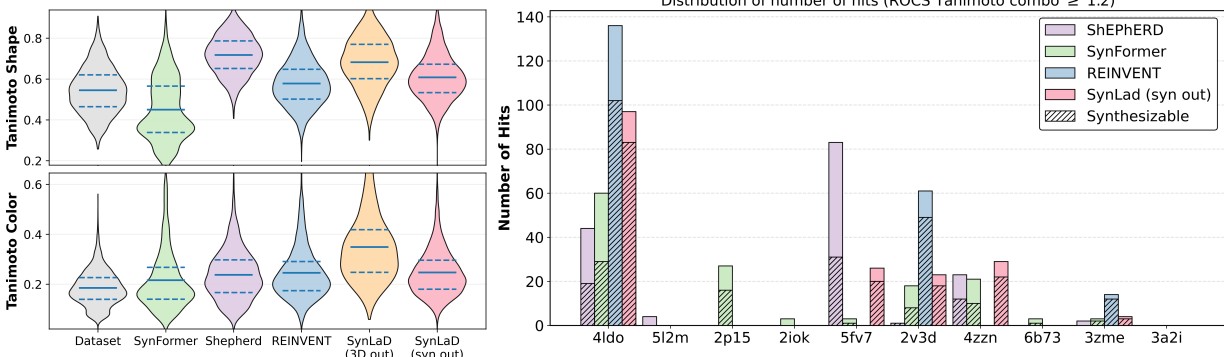

*Figure 4.* **Bioactive hit diversification.** Structural analogues are generated for 10 Lit-PCBA ligands using SynLaD and baseline methods. We report (1) the distribution of aggregated Tanimoto shape and color similarity scores to the query, and (2) the number of hits per query for each method, highlighting the fraction of synthesizable hits.

in average number of synthesizable hits.

Although REINVENT achieves the highest average number of synthesizable hits, this result is dominated by a single query with an unusually large hit count; across the benchmark, it generates synthesizable hits for only three queries and shows limited scaffold diversity. We show examples of query-generated molecule pairs with highlighted pharmacophore and shape overlap in Figure 5 and further examples of synthesis and 3D decoder outputs in Appendix Figure 9.

While we report results for both decoders for completeness, the key strength of our approach lies in the synthesis decoder, which—despite operating on linear synthesis trajectories—learns to satisfy shape and pharmacophore constraints more effectively than baselines that do not learn 3D information. This highlights the benefit of our strategy: learning latent representations that encode rich 3D conditioning signals and using them to guide synthesis generation. We further support this claim in Appendix C.3, where we perform two ablations: 1) we remove the structure-aware latent component (3D reconstruction heads and diffusion module) from SynLaD and keep the synthesis architecture and pharmacophore-conditioning fixed and 2) we switch the 3D reconstruction task for SMILES reconstruction, and keep the rest of the framework identical. We see that both ablations lead to a substantial performance drop, demon-

strating that the structure-aware latent component is critical.

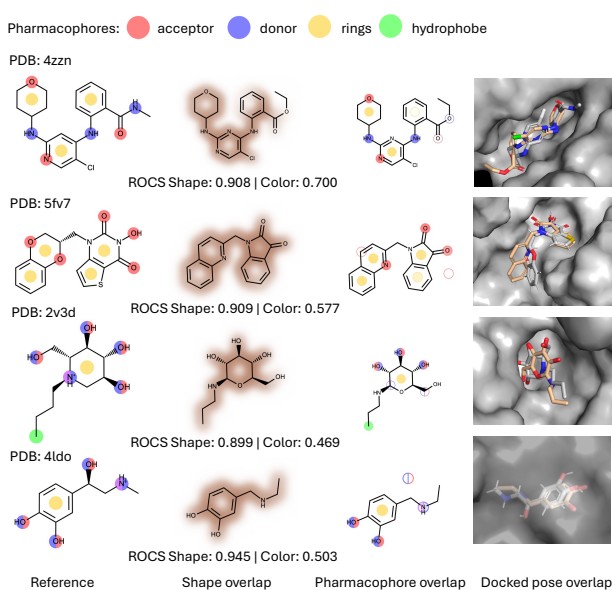

*Figure 5.* **Bioactive hit diversification experiment**. Examples of molecules generated by our model with annotated shape and pharmacophore overlap. The right-hand side shows the native ligand (gray) overlaid with the docked generated molecule (orange).

## 5. Conclusion

In this work, we introduce a conditional latent diffusion framework that decodes latent representations into both 3D molecular structures and reaction-based synthesis pathways. We show that jointly training these two decoders is mutually beneficial at sampling time: it increases the synthesizability of molecules produced by the 3D decoder and improves the synthesis decoder's ability to satisfy shape/pharmacophore constraints under conditioning, while also enchancing sample diversity. Overall, our results demonstrate that conditioning an autoregressive synthesis decoder through a 3D-informed latent representation yields high numbers of diverse, synthesizable, pharmacophore-aligned hits. Looking ahead, an exciting direction is to scale our method to larger datasets (e.g., Pistachio, NextMove Software, 2025), which would enable access to more novel transformations and more combinatorially complex regions of chemical space.

## Impact Statement

This paper presents a generative modeling approach aimed at advancing small-molecule drug discovery by jointly addressing 3D design objectives and synthetic feasibility. By enabling the generation of molecules that are both pharmacophore-consistent and synthesizable, our work has the potential to accelerate early-stage drug discovery and reduce reliance on costly and time-consuming experimental screening pipelines.

At the same time, we acknowledge that generative models for molecular design could, in principle, be misused to propose harmful or toxic compounds. While our work does not explicitly target such applications and is focused on drug discovery settings, the underlying methodology could be adapted in unintended ways.

We emphasize that our approach operates within standard drug discovery pipelines, where generated candidates are subject to extensive downstream validation, including safety, synthesizability, and biological evaluation. As such, we believe the primary impact of this work is to support responsible and beneficial applications in medicinal chemistry.

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

H. SynCoGen: Synthesizable 3D molecule generation
via joint reaction and coordinate modeling, 2025. URL
https://arxiv.org/abs/2507.11818.

Rezende, D. and Mohamed, S. Variational inference with
normalizing flows. In Bach, F. and Blei, D. (eds.), *Pro-
ceedings of the 32nd International Conference on Ma-
chine Learning*, volume 37 of *Proceedings of Machine
Learning Research*, pp. 1530–1538, Lille, France, 07–09
Jul 2015. PMLR. URL https://proceedings.ml
r.press/v37/rezende15.html.

Rombach, R., Blattmann, A., Lorenz, D., Esser, P., and
Ommer, B. High-resolution image synthesis with latent
diffusion models. In *2022 IEEE/CVF Conference on
Computer Vision and Pattern Recognition (CVPR)*, pp.
10674–10685, Los Alamitos, CA, USA, June 2022. IEEE
Computer Society. URL https://doi.ieeecomp
utersociety.org/10.1109/CVPR52688.20
22.01042.

Sacha, M., Błaz, M., Byrski, P., Dabrowski-Tumanski, P.,
Chrominski, M., Loska, R., Włodarczyk-Pruszynski, P.,
and Jastrzebski, S. Molecule edit graph attention network:
Modeling chemical reactions as sequences of graph edits.
*Journal of Chemical Information and Modeling*, 61(7):
3273–3284, 2021. URL https://doi.org/10.1
021/acs.jcim.1c00537.

Sagawa, T. and Kojima, R. ReactionT5: a pre-trained trans-
former model for accurate chemical reaction prediction
with limited data. *Journal of Cheminformatics*, 17(1):
126, 2025. URL https://doi.org/10.1186/s1
3321-025-01075-4.

Schneuing, A., Harris, C., Du, Y., Didi, K., Jamasb, A.,
Igashov, I., Du, W., Gomes, C., Blundell, T. L., Lio, P.,
Welling, M., Bronstein, M., and Correia, B. Structure-
based drug design with equivariant diffusion models. *Na-
ture Computational Science*, 4(12):899–909, 2024. URL
https://doi.org/10.1038/s43588-024-0
0737-x.

Schneuing, A., Igashov, I., Dobbelstein, A. W., Castiglione,
T., Bronstein, M. M., and Correia, B. Multi-domain distri-
bution learning for de novo drug design. In *The Thirteenth
International Conference on Learning Representations*,
2025. URL https://openreview.net/forum
?id=g3VCIM94ke.

Schwaller, P., Gaudin, T., Lanyi, D., Bekas, C., and Laino, T.
"Found in Translation": predicting outcomes of complex
organic chemistry reactions using neural sequence-to-
sequence models. *Chemical Science*, 9(28):6091–6098,
2018. URL https://doi.org/10.1039/C8SC
02339E.

Schwaller, P., Laino, T., Gaudin, T., Bolgar, P., Hunter,
C. A., Bekas, C., and Lee, A. A. Molecular Transformer:
A model for uncertainty-calibrated chemical reaction pre-
diction. *ACS Central Science*, 5(9):1572–1583, 2019.
URL https://doi.org/10.1021/acscents
ci.9b00576.

Segler, M. H., Kogej, T., Tyrchan, C., and Waller, M. P.
Generating focused molecule libraries for drug discovery
with recurrent neural networks. *ACS Central Science*, 4
(1):120–131, 2018. URL https://doi.org/10.1
021/acscentsci.7b00512.

Seidl, P., Renz, P., Dyubankova, N., Neves, P., Verhoeven,
J., Wegner, J. K., Segler, M., Hochreiter, S., and Klam-
bauer, G. Improving few-and zero-shot reaction template
prediction using modern hopfield networks. *Journal of
Chemical Information and Modeling*, 62(9):2111–2120,
2022. URL https://doi.org/10.1021/acs.
jcim.1c01065.

Seo, S., Kim, M., Shen, T., Ester, M., Park, J., Ahn, S., and
Kim, W. Y. Generative flows on synthetic pathway for
drug design. In *The Thirteenth International Conference
on Learning Representations*, 2025. URL https://op
enreview.net/forum?id=pB1XSj2y4X.

Shen, T., Seo, S., Irwin, R., Didi, K., Olsson, S., Kim, W. Y.,
and Ester, M. Compositional flows for 3D molecule and

synthesis pathway co-design. In *Forty-second International Conference on Machine Learning*, 2025. URL https://openreview.net/forum?id=4aXfSLfM0Z.

Sheridan, R. P., McGaughey, G. B., and Cornell, W. D. Multiple protein structures and multiple ligands: effects on the apparent goodness of virtual screening results. *Journal of Computer-Aided Molecular Design*, 22(3):257–265, 2008. URL https://doi.org/10.1007/s10822-008-9168-9.

Sohl-Dickstein, J., Weiss, E., Maheswaranathan, N., and Ganguli, S. Deep unsupervised learning using nonequilibrium thermodynamics. In Bach, F. and Blei, D. (eds.), *Proceedings of the 32nd International Conference on Machine Learning*, volume 37 of *Proceedings of Machine Learning Research*, pp. 2256–2265, Lille, France, 07–09 Jul 2015. PMLR. URL https://proceedings.mlr.press/v37/sohl-dickstein15.html.

Song, Y. and Ermon, S. Generative modeling by estimating gradients of the data distribution. In Wallach, H., Larochelle, H., Beygelzimer, A., d'Alché-Buc, F., Fox, E., and Garnett, R. (eds.), *Advances in Neural Information Processing Systems*, volume 32. Curran Associates, Inc., 2019. URL https://proceedings.neurips.cc/paper_files/paper/2019/file/3001ef257407d5a371a96dcd947c7d93-Paper.pdf.

Stanley, M. and Segler, M. Fake it until you make it? Generative de novo design and virtual screening of synthesizable molecules. *Current Opinion in Structural Biology*, 82:102658, 2023. URL https://doi.org/10.1016/j.sbi.2023.102658.

Sutskever, I., Vinyals, O., and Le, Q. V. Sequence to sequence learning with neural networks. In Ghahramani, Z., Welling, M., Cortes, C., Lawrence, N., and Weinberger, K. (eds.), *Advances in Neural Information Processing Systems*, volume 27. Curran Associates, Inc., 2014. URL https://proceedings.neurips.cc/paper_files/paper/2014/file/5a18e133cbf9f257297f410bb7eca942-Paper.pdf.

Swanson, K., Liu, G., Catacutan, D. B., Arnold, A., Zou, J., and Stokes, J. M. Generative AI for designing and validating easily synthesizable and structurally novel antibiotics. *Nature Machine Intelligence*, 6(3):338–353, 2024. URL https://doi.org/10.1038/s42256-024-00809-7.

Swanson, K., Liu, G., Catacutan, D. B., McLellan, S., Arnold, A., Tu, M. M., Brown, E. D., Zou, J., and Stokes, J. M. SyntheMol-RL: a flexible reinforcement learning framework for designing novel and synthesizable antibiotics. *bioRxiv*, 2025. URL https://doi.org/10.1101/2025.05.17.654017.

Tran-Nguyen, V.-K., Jacquemard, C., and Rognan, D. LIT-PCBA: An unbiased data set for machine learning and virtual screening. *Journal of Chemical Information and Modeling*, 60(9):4263–4273, 2020. URL https://doi.org/10.1021/acs.jcim.0c00155.

Tripp, A., Daxberger, E., and Hernández-Lobato, J. M. Sample-efficient optimization in the latent space of deep generative models via weighted retraining. In Larochelle, H., Ranzato, M., Hadsell, R., Balcan, M., and Lin, H. (eds.), *Advances in Neural Information Processing Systems*, volume 33, pp. 11259–11272. Curran Associates, Inc., 2020. URL https://proceedings.neurips.cc/paper_files/paper/2020/file/81e3225c6ad49623167a4309eb4b2e75-Paper.pdf.

Tu, Z. and Coley, C. W. Permutation invariant graph-to-sequence model for template-free retrosynthesis and reaction prediction. *Journal of Chemical Information and Modeling*, 62(15):3503–3513, 2022. URL https://doi.org/10.1021/acs.jcim.2c00321.

Vahdat, A., Kreis, K., and Kautz, J. Score-based generative modeling in latent space. In Ranzato, M., Beygelzimer, A., Dauphin, Y., Liang, P., and Vaughan, J. W. (eds.), *Advances in Neural Information Processing Systems*, volume 34, pp. 11287–11302. Curran Associates, Inc., 2021. URL https://proceedings.neurips.cc/paper_files/paper/2021/file/5dca4c6b9e244d24a30b4c45601d9720-Paper.pdf.

Vaswani, A., Shazeer, N., Parmar, N., Uszkoreit, J., Jones, L., Gomez, A. N., Kaiser, L., and Polosukhin, I. Attention is all you need. In Guyon, I., Luxburg, U. V., Bengio, S., Wallach, H., Fergus, R., Vishwanathan, S., and Garnett, R. (eds.), *Advances in Neural Information Processing Systems*, volume 30. Curran Associates, Inc., 2017. URL https://proceedings.neurips.cc/paper_files/paper/2017/file/3f5ee243547dee91fbd053c1c4a845aa-Paper.pdf.

Vinkers, H. M., Jonge, M. R. d., Daeyaert, F. F. D., Heeres, J., Koymans, L. M. H., Lenthe, J. H. v., Lewi, P. J., Timmerman, H., Aken, K. V., and Janssen, P. A. J. SYNOPSIS: SYNthesize and OPtimize system in silico. *Journal of Medicinal Chemistry*, 46(13):2765–2773, 2003. URL https://doi.org/10.1021/jm030809x.

Vonessen, C., Harris, C., Cretu, M., and Liò, P. Tabasco: A fast, simplified model for molecular generation with improved physical quality, 2025. URL https://arxiv.org/abs/2507.00899.

Wei, J. N., Duvenaud, D., and Aspuru-Guzik, A. Neural networks for the prediction of organic chemistry reactions. *ACS Central Science*, 2(10):725–732, 2016. URL https://doi.org/10.1021/acscentsci.6b00219.

Wolf, T., Debut, L., Sanh, V., Chaumond, J., Delangue, C., Moi, A., Cistac, P., Rault, T., Louf, R., Funtowicz, M., Davison, J., Shleifer, S., von Platen, P., Ma, C., Jernite, Y., Plu, J., Xu, C., Le Scao, T., Gugger, S., Drame, M., Lhoest, Q., and Rush, A. Transformers: State-of-the-art natural language processing. In Liu, Q. and Schlangen, D. (eds.), *Proceedings of the 2020 Conference on Empirical Methods in Natural Language Processing: System Demonstrations*, pp. 38–45, Online, October 2020. Association for Computational Linguistics. URL https://aclanthology.org/2020.emnlp-demos.6/.

Xie, W., Zhang, J., Xie, Q., Gong, C., Ren, Y., Xie, J., Sun, Q., Xu, Y., Lai, L., and Pei, J. Accelerating discovery of bioactive ligands with pharmacophore-informed generative models. *Nature Communications*, 16(1):2391, 2025. URL https://doi.org/10.1038/s41467-025-56349-0.

Xu, M., Powers, A., Dror, R., Ermon, S., and Leskovec, J. Geometric latent diffusion models for 3D molecule generation, 2023. URL https://arxiv.org/abs/2305.01140.

Yim, J., Trippe, B. L., De Bortoli, V., Mathieu, E., Doucet, A., Barzilay, R., and Jaakkola, T. SE(3) diffusion model with application to protein backbone generation. In *Proceedings of the 40th International Conference on Machine Learning*, ICML'23. JMLR.org, 2023. URL https://dl.acm.org/doi/10.5555/3618408.3620080.

Yu, K., Roh, J., Li, Z., Gao, W., Wang, R., and Coley, C. W. Double-ended synthesis planning with goal-constrained bidirectional search. In Globerson, A., Mackey, L., Belgrave, D., Fan, A., Paquet, U., Tomczak, J., and Zhang, C. (eds.), *Advances in Neural Information Processing Systems*, volume 37, pp. 112919–112949. Curran Associates, Inc., 2024. URL https://doi.org/10.52202/079017-3588.

Zhu, H., Zhou, R., Cao, D., Tang, J., and Li, M. A pharmacophore-guided deep learning approach for bioactive molecular generation. *Nature Communications*, 14(1):6234, 2023. URL https://doi.org/10.1038/s41467-023-41454-9.

Ziv, Y., Imrie, F., Marsden, B., and Deane, C. M. MolSnapper: Conditioning diffusion for structure-based drug design. *Journal of Chemical Information and Modeling*, 65(9):4263–4273, 2025. URL https://doi.org/10.1021/acs.jcim.4c02008.

# A. Limitations and Future Work

The focus of this work is to introduce and validate SynLaD's ability to generate small molecules that preserve key phramacophore interactions of ligands and have high synthesizability. To this end, we restrict our methodology and experiments to ligand-based pharmacophore conditioning, reaction-constrained synthesis-pathway generation, and evaluation on small-molecule analogue generation tasks, as well as evaluation of our method's core modules. A current limitation of our framework is that it relies on external reaction predicitons, and its performance is therefore affected by predictor bias or failure models. Future work could evaluate alternative predictors or ensembles of predictors. Second, our experiments are limited to the dataset scales considered in this work, and scaling to larger and more diverse reaction datasets may further test the generality of the approach. Future work could also focus on improving cross-decoder agreement, as stronger consistency across decoding heads may lead to better molecule feature alignment.

# B. Methods

## B.1. Dataset

For our dataset, we make use of the provided synthesis DAGs from Bradshaw et al. (2020). Briefly, these DAGs were obtained from a cleaned version of the USPTO reaction dataset after stripping out reagents (Jin et al., 2017; Lowe, 2017); first, by building up a reaction network from the USPTO reactions and an initial set of "building block" nodes (picked by frequency), before extracting a possible synthesis plan for each non-building block node by tracing reactions backward to a loop-free subgraph that terminates in building blocks (selecting a single route when multiple alternatives exist). Using this data, we first filter the reactions such that the final products contain between 10 and 40 heavy atoms, and obtain $67\,512$ synthesis DAGs, 90% of which we keep as our training set and use the remaining data for our validation and test sets. Note that we randomly split by final molecule (i.e., the different train, validation, and test sets lead to different final molecules but may share reactions/sub-networks); analyzing SynLaD's ability to extrapolate to different data splits (for instance, a "Reachable" vs "Hard" split akin to Yu et al., 2024, p.7) is an interesting future direction to explore. For each final product of the synthesis DAGs we compute a maximum of 10 low-energy conformers using Omega (Hawkins et al., 2010), which results in a total of $584\,896$ 3D molecules.

## B.2. Reaction network serialization

A synthesis pathway here is defined as a directed acyclic graph (DAG), where nodes represent molecules (each unique molecule maps to a unique node) and the edges represent reactions. In order to describe these pathways using an autoregressive language model we need to serialize them into a linear sequence of tokens. Our proposed approach follows (Bradshaw et al., 2020) (alternative serialization schemes have also been proposed, Gao et al., 2025; Lee et al., 2026; Luo et al., 2024) and is shown in Figure 6. The scheme relies on two kinds of tokens: *action tokens* and *molecular tokens*. Action tokens define a new operation to take with respect to the graph and are generally followed by a series of molecular nodes that specify the arguments required to define the action's effect. There are three types of action tokens:

- `B` adds a building block node to the network (these are easily purchasable compounds and the token is followed by a single molecule token defining the identity of the chosen building block).

- `F` adds a reaction edge (and a product node if the molecule does not yet exist in the network) to our network through a single forward reaction. This action is followed by selecting one or more molecule nodes (representing the existing molecules in the network) to use as reactants, a sub-action `F_r` (which indicates the reaction should be run) and a final product node (followed by `P`) that represents the product of the reaction.

- `S` is the stop action, which halts the prediction.

## B.3. Synthesis action embeddings

The input tokens to the transformer are obtained via summing a series of embeddings as shown in Figure 1C: a learned embedding which specifies the token type, a molecule embedding (if the action involves a molecule) which is a projection of the molecule's 2048-bit Morgan fingerprint, and a learned positional embedding.

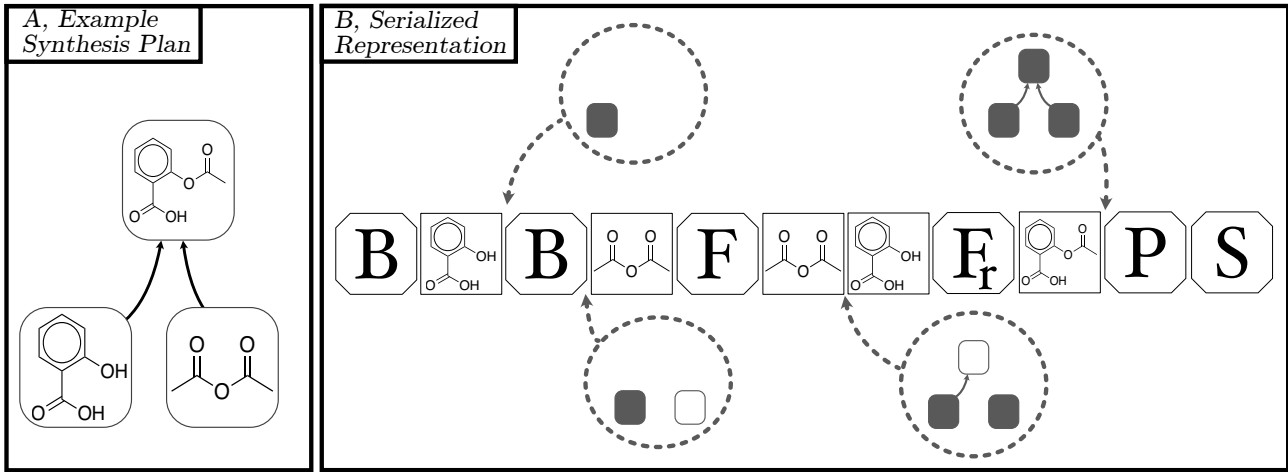

*Figure 6.* A tokenization scheme for describing synthesis plans in a bottom-up manner similar to Bradshaw et al. (2020). The synthesis plan, depicted as a DAG in **A** can be serialized as shown in **B** into a series of tokens. In **B**, the four dotted circles above the sequence indicate the state of the synthesis plan at that stage of decoding—empty nodes indicate the identity of the molecule corresponding to that node has yet to be defined. Note that this scheme can be used to describe complex, multi-step synthesis plans in a bottom-up manner.

*Table 5.* **Autoencoder loss weights ablation study.** We report 3D molecule match rate (computed with MoleculeMatcher from PyMatGen) and invariant building block match rate in the synthesis sequence.

| $\lambda$ | $\beta$ | 3D molecule match rate | Building block match rate |
|---|---|---|---|
| 0.50 | $10^{-5}$ | 82.1 | 83.3 |
| 0.70 | $10^{-5}$ | 90.2 | 83.1 |
| 0.92 | $10^{-5}$ | 98.5 | 82.5 |
| 0.98 | $10^{-5}$ | 98.7 | 80.6 |
| 0.92 | $10^{-4}$ | 80.6 | 67.4 |

## B.4. VAE training

**Autoencoder ablations**   We report ablations for autoencoder hyperparameters in Tables 5 and 6. Increasing the weight on the synthesis decoder loss (see Eq. (1)) harms 3D coordinate reconstruction and adds little benefit for the synthesis sequence reconstruction. Note that we report the accuracy of builiding block prediction in its order-invariant form (i.e., building blocks might be added earlier or later on in the sequence, with the end product being the same); this differs during training in which we randomly pick an order (for building blocks at the same level) and train on this fixed sequence using teacher forcing. We use $\lambda = 0.92$ in our experiments and $\beta = 10^{-5}$, as larger KL weight harms reconstruction accuracy for both decoders. We use a latent dimension $d = 16$ which shows the best molecule and synthesis reconstruction.

### B.4.1. SYNTHESIS DECODER TRAINING

The synthesis decoder is implemented as a decoder-only transformer. Each input token is embedded by summing (i) a learned embedding for special tokens or retrieved molecular embeddings, (ii) a learned token-type embedding, and (iii) a learned positional embedding. Graph-level information is provided via a fingerprint encoder. The embedded sequence is processed by 12 transformer blocks with RMSNorm pre-normalization, multi-head self-attention, and a SwiGLU feed-forward network (further details can be found in our attached code). When conditioning is enabled, each block additionally applies a cross-attention sublayer over an external conditioning vector. During training, we use teacher forcing (Goodfellow et al., 2016, §10.2.1), i.e., for each prediction step the decoder is conditioned on the ground-truth previous tokens. We compute the cross-entropy loss only on positions where the model is expected to generate tokens, and mask out product tokens (and any padded positions) so they do not contribute to the loss. We report ablations for synthesis decoder inference in Tables 7 and 8.

*Table 6.* **Autoencoder reconstruction accuracies.** The synthesis match rate and Morgan fingerprint Tanimoto similarity are calculated between the final product of the predicted and the ground truth synthesis sequences. We report performance with two different inference strategies for the synthesis decoder.

| Autoencoder hyperparameters | | 3D metrics | | Synthesis metrics | |
|---|---|---|---|---|---|
| Inference method | Latent dim. | Match Rate (%)↑ | RMSD (Å) ↓ | Match Rate (%)↑ | Tanimoto similarity↑ |
| Beam search | 8 | 98.9 | 0.05 | 58.4 | 0.82 |
| Beam search | 16 | 98.5 | 0.05 | 63.4 | 0.84 |
| Beam search | 32 | 98.9 | 0.04 | 62.7 | 0.83 |
| Sampling | 8 | 98.9 | 0.05 | 42.8 | 0.73 |
| Sampling | 16 | 98.5 | 0.05 | 49.0 | 0.76 |
| Sampling | 32 | 98.5 | 0.04 | 51.4 | 0.77 |

*Table 7.* **Synthesis decoder ablations (beam search).**

| Beam width | Synthesis Match Rate (%)↑ | Tanimoto similarity↑ |
|---|---|---|
| 1 | 47.6 | 0.75 |
| 5 | 63.4 | 0.84 |
| 10 | 59.5 | 0.82 |

*Table 8.* **Synthesis decoder ablations (sampling).**

| N | Top-$k$ | Synthesis Match Rate (%)↑ | Tanimoto similarity↑ |
|---|---|---|---|
| 1 | 1 | 43.9 | 0.73 |
| 1 | 5 | 44.1 | 0.73 |
| 1 | 10 | 43.4 | 0.73 |

## B.5. DiT training

**Sampling ablations.** We investigate the effect of the number of sampling steps used during inference on molecule quality and synthesis/3D decoder output match rate in Table 9. Unless otherwise stated, we use 100 sampling steps in all experiments. We additionally ablate the classifier-free guidance weight $w$ in Eq. (5) (Table 10). Increasing $w$ improves control over latent quality and yields more valid decoded molecules, but reduces scaffold diversity. Since our primary objective is to maximize the number of generated hits, we therefore use $w = 0$ (i.e., no classifier-free guidance) throughout.

## B.6. Training and hyperparameters

We jointly train the VAE and synthesis decoder, then train the DiT in a second stage, using AdamW with a constant learning rate of $10^{-4}$, zero weight decay, and a batch size of 256. We maintain an exponential moving average of the DiT parameters with decay 0.9999. Each stage is run to convergence, for up to 3000 epochs or 3 days on a single B200 GPU.

For the VAE, we use standard transformers for both the encoder and 3D decoder with hidden dimension $d_{\text{model}} = 256$, 4 attention heads and 6 layers (9.5M parameters total). For the synthesis decoder we used $d_{\text{model}} = 512$, 16 attention heads, and 12 layers with causal self-attention, cross-attention conditioning to the latents, and SwigLU FFNs (65M parameters total).

For the second stage training, the pharmacophore encoder is a standard transformer with $d_{\text{model}} = 256$, 4 heads, and 4 layers (40M parameters total), and the DiT denoiser has $d_{\text{model}} = 768$, 12 attention heads, and 12 layers (130M parameters total).

**Molecule and pharmacophore embeddings.** Molecules and pharmacophores are represented by their respective atom or pharmacophore types together with 3D coordinates (see Sections 3.1 and 3.2). We use separate embedding layers for atom /

*Table 9.* **Number of sampling steps ablation.** We performed the ablation in the unconditional setting.

| #Steps | Validity↑ | PoseBusters↑ | Synthesis/3D exact match rate↑ |
|---|---|---|---|
| 50 | 0.85 | 0.85 | 0.28 |
| 100 | 0.89 | 0.89 | 0.33 |
| 200 | 0.89 | 0.89 | 0.32 |
| 300 | 0.90 | 0.89 | 0.33 |
| 500 | 0.91 | 0.89 | 0.33 |

*Table 10.* **Classifier free guidance ablation.** We performed the ablation for the Lit-PCBA ligands conditioning experiment and generated 100 samples per conditioning input.

| cfg. setting | Validity↑ | Diversity↑ | | Hits↑ | | Uniq. scaff. hits↑ | |
|---|---|---|---|---|---|---|---|
| | | (syn out) | (3D out) | (syn out) | (3D out) | (syn out) | (3D out) |
| w/o cfg. | 0.49 | 0.86 | 0.86 | 17.9 | 38.1 | 6.9 | 18.8 |
| $w = 0.5$ | 0.56 | 0.85 | 0.85 | 14.2 | 34.7 | 3.8 | 12.1 |
| $w = 1.0$ | 0.46 | 0.85 | 0.84 | 14.3 | 38.0 | 4.8 | 15.2 |
| $w = 2.0$ | 0.45 | 0.85 | 0.84 | 13.1 | 34.7 | 5.0 | 15.7 |
| $w = 4.0$ | 0.36 | 0.85 | 0.84 | 11.7 | 23.4 | 5.2 | 12.6 |

*Table 11.* **Reaction predictor test accuracy.** Top-k accuracy (%) on the held-out test set for product outcome prediction.

| Checkpoint | Top-1 | Top-2 | Top-3 | Top-5 |
|---|---|---|---|---|
| Final (1M iters) | 84.6 | 91.4 | 93.3 | 94.8 |

pharmacophore types and atomic positions, following Joshi et al. (2025). Atom types are embedded via a learned embedding lookup, while positions are projected using a linear layer. The resulting representations are combined by summation.

### B.7. Reaction predictor oracle training

As explained in the main text, when running our synthesis decoder at inference time we require access to a reaction predictor oracle to predict the molecular identity of the product molecule given the set of reactants (we ignore conditions and reagents in this current work). For this, we use a model based around the BART architecture (Lewis et al., 2020), using an implementation adapted from the `transformers` library (Wolf et al., 2020) for use in reaction prediction (Bradshaw et al., 2025). This is similar in spirit to other encoder-decoder models used for reaction prediction/one-step retrosynthesis (Nam & Kim, 2016; Schwaller et al., 2019; 2018; Liu et al., 2017). We train our reaction prediction oracle on reactant-product pairs from the same USPTO dataset we extract the synthesis plans from for training SynLaD; teacher forcing is used at training time and beam search is used during inference for making predictions. We leave to future work the exploration of using alternative reaction predictors (Tu & Coley, 2022; Jin et al., 2017; Bradshaw et al., 2019a; Seidl et al., 2022; Joung et al., 2025; Fooshee et al., 2018; Kayala et al., 2011; Sacha et al., 2021; Wei et al., 2016; Do et al., 2019; Bi et al., 2021; Irwin et al., 2022; Sagawa & Kojima, 2025), or even multiple at a time (Maziarz et al., 2025) for more robust predictions. We report accuracy results from training on a train/validation/test split of approximately 409k/30k/40k data points extracted from the USPTO dataset in Table 11.

### B.8. Evaluation metrics

We define the metrics used thoroughly in the paper as follows:

- **Validity (Valid.)**: % of molecules that can be processed by RDKit.

- **Internal diversity (IntDiv.)**: One minus the value of the averaged pairwise Tanimoto similarities of the Morgan fingerprints (size 2048, radius 2) of the molecules in the generated set.

- **Murcko Scaffold count**: number of unique Murcko scaffolds in a generated set.

- **Novelty (Nov.)**: % of valid molecules not present in the training set.

- **PoseBusters (PB)**: % of molecules passing all PoseBusters filters (Buttenschoen et al., 2024). This was only evaluated for models that generated 3D poses.

- **FCD** (Fréchet ChemNet Distance): measures distributional similarity to a reference dataset using ChemNet embeddings (lower is better) (Preuer et al., 2018).

- **SA**: Averaged synthetic accessibility score from Ertl & Schuffenhauer (2009).

- **AiZynthFinder (AiZynth.)**: % of molecules (out of 100) for which AiZynthFinder finds valid synthetic pathways given its default set of reactions and ZINC building blocks.

- **RMSD and match rate**: For 3D molecule reconstruction we report RMSD and match rate computed using MoleculeMatcher from PyMatGen (Ong et al., 2013).

- **Hit rate**: We compute hit rate by counting the number of unique molecules with ROCS Tanimoto Combo score $\geq 1.2$.

### B.9. ROCS evaluation pipeline

We scored 3D similarity with OpenEye ROCS (Hawkins et al., 2007) using a standard OMEGA→ROCS workflow. We kept the query conformation intact (from our test set or Lit-PCBA benchmark pose) and for each candidate molecule, we first generated an ensemble of 100 low-energy 3D conformations with OMEGA (Hawkins et al., 2010), which constructs initial geometries from fragment conformers and then samples torsions using a knowledge-based torsion library, retaining a pruned set of energetically and geometrically reasonable conformers. We then used ROCS to perform fast rigid overlays between all query-candidate conformer pairs, representing molecular shape as a sum of atom-centered Gaussian functions and maximizing shared volume and pharmacophoric ("color") feature overlap. For each candidate, we reported the best overlay across conformers, and computed ShapeTanimoto, ColorTanimoto, and their sum (TanimotoCombo) as the final shape, color, and combined similarity scores. The same pipeline was applied for SynLaD and all baselines, regardless of the methods' output types.

## C. Experiments

### C.1. Pharmacophore conditioned generation

During training, we load the pharmacophore features for a conformer and randomly drop $k$ features, where $k$ is sampled uniformly from $[0, \max(0, N_{ph} - 3)]$, and $N_{ph}$ denotes the number of pharmacophore features in the sample. During sampling, we set the latent to contain $M$ atoms, with $M$ sampled uniformly from $[\max(1, N - 3), N + 3]$, where $N$ is the number of heavy atoms in the query (i.e., excluding hydrogens).

*Table 12.* **Molecule quality metrics for in-distribution pharmacophore conditioning.** Higher is better for all.

| Method | Validity | Uniqueness | Diversity |
|---|---|---|---|
| SynLaD (3D out) | 0.78 | 0.60 | 0.72 |
| SynLaD (syn out) | 1.0 | 0.63 | 0.75 |

### C.2. Bioactive hit diversification experiment

We used ten PDB targets from Tran-Nguyen et al. (2020), with PDB IDs 4LDO, 5L2M, 2P15, 2IOK, 5FV7, 2V3D, 4ZZN, 6B73, 3ZME, 3A2I. We extracted the corresponding ligand for each of these as the 'query'. We show examples of decoded outputs, together with the conditioning query in Figure 9. We show additional results (including the per-query number of hits for the 3D decoder) in Figure 11.

For the hit diversification experiment, we quantify the extent to which molecules are out-of-distribution from the training set by plotting Tanimoto distances of the Lit-PCBA ligands to all training molecules in Figure 12 (computed using 2048-bit Morgan fingerprints).

In addition to the results in Table 4, we provide a ranking table that summarizes the relative performance of each method across all evaluated metrics. For each metric, methods are ranked per task and the average rank is reported, using a joint ranking scheme. This view highlights the consistency of each method across objectives, rather than emphasizing performance on any single metric. As shown in Table 13, SynLaD achieves the best or tied-best average rank for validity, unique scaffold hits, and synthesizable hits.

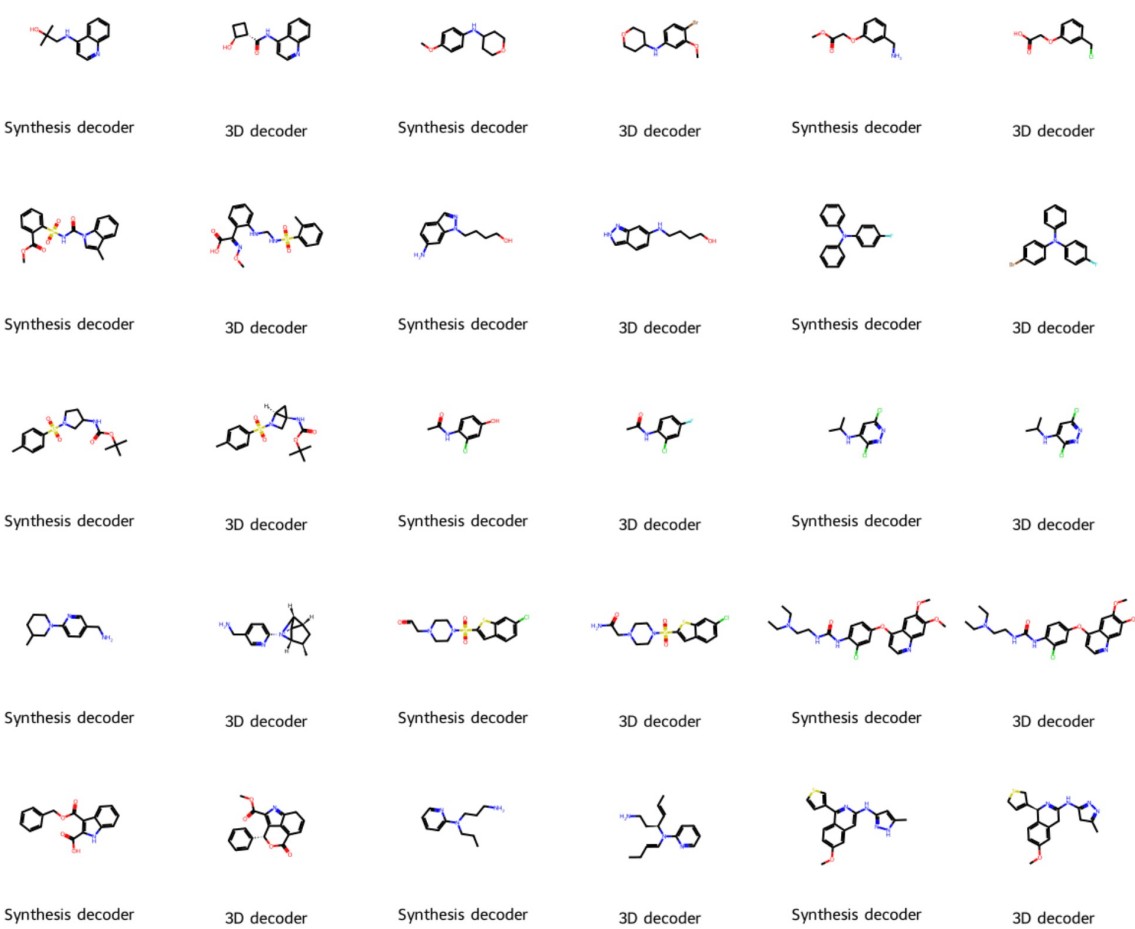

*Figure 7.* **Unconditional generation: synthesis- vs. 3D-decoded outputs.** Examples of decoded outputs from the synthesis decoder, followed by outputs of the 3D decoder, from the same latent.

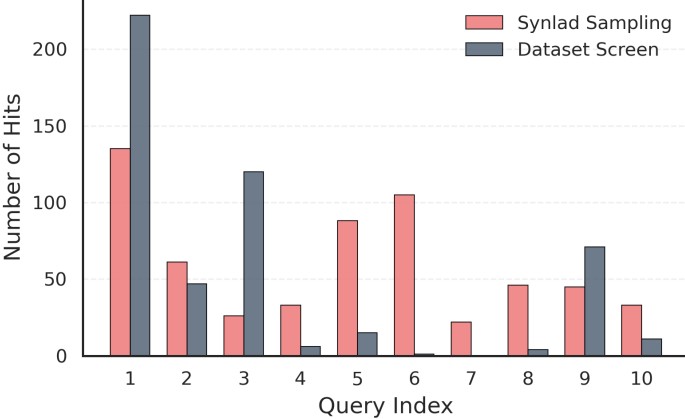

*Figure 8.* **Dataset screen vs. SynLaD sampling**. We report the number of unique hits for randomly selected queries from the test set. See the "Screening case study" section in the main text for further details.

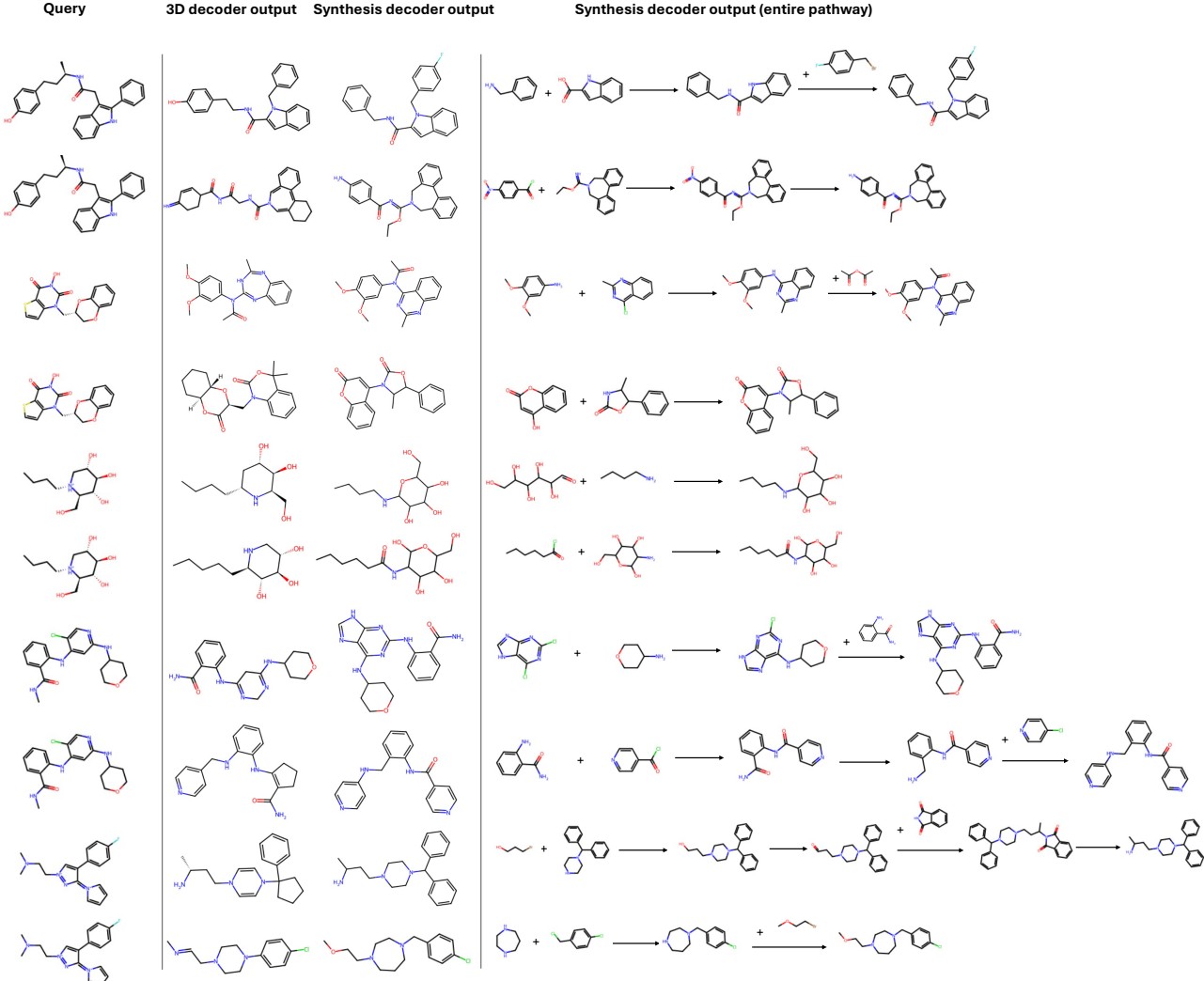

*Figure 9.* **SynLaD outputs for the Lit-PCBA conditioning task.** We visualize the Lit-PCBA ligand used as the conditioning **Query** (via its pharmacophore), together with the corresponding **3D decoder output** and the **synthesis decoder output** (final product and **entire pathway**). For readability, the query and 3D-decoded molecules are shown in 2D, although both are modeled in 3D coordinate space. We intentionally select examples where the synthesis- and 3D-decoder outputs do not coincide.

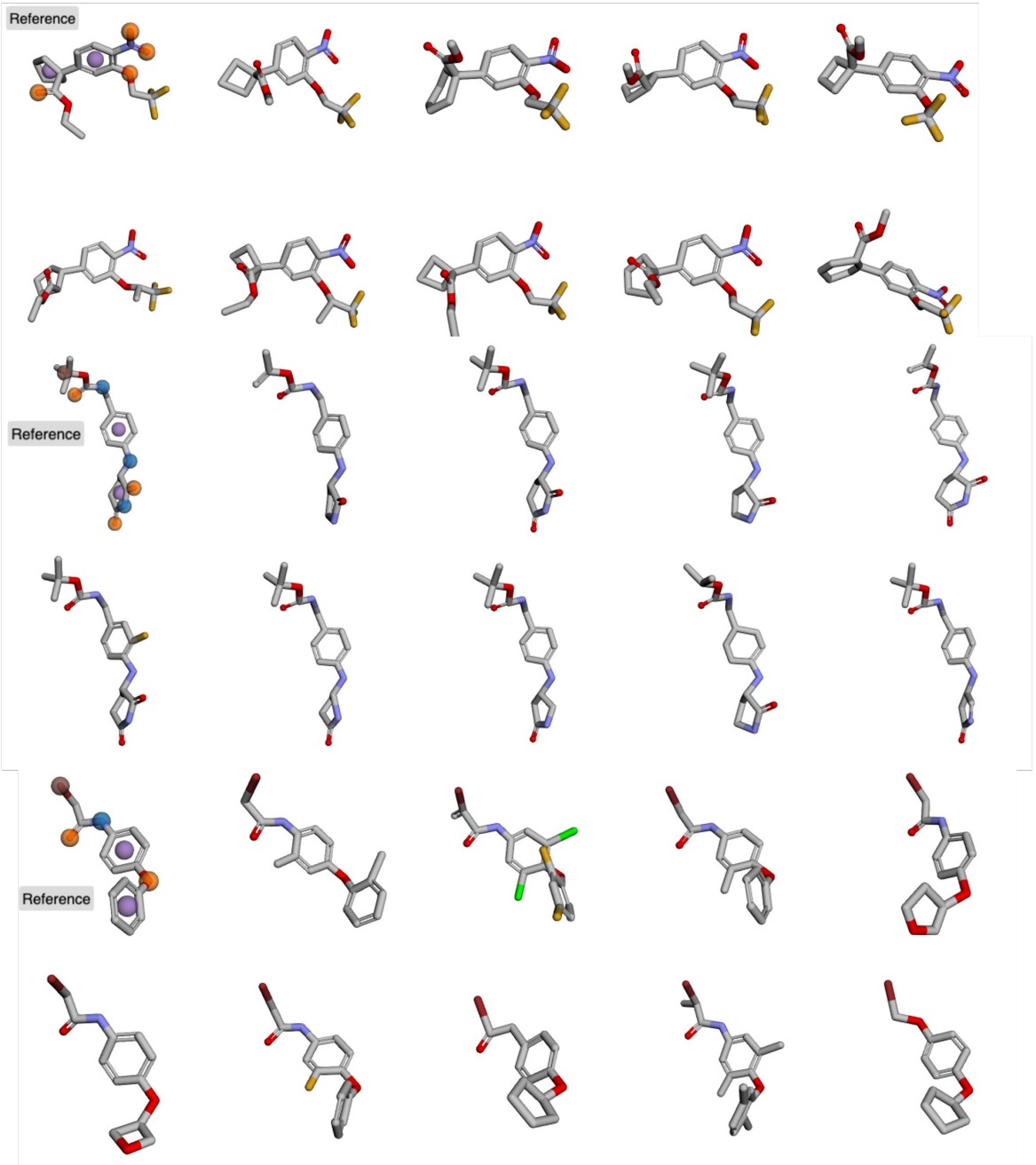

*Figure 10.* **Examples of query (reference) and sampled molecules conditioned on pharmacophores.** Samples were decoded using the 3D decoder, and we show the molecules in their predicted conformation.

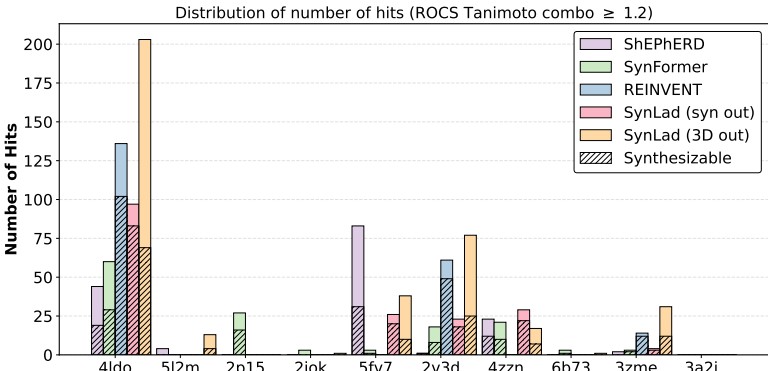

*Figure 11.* **Bioactive hit difersification experiment.** We add to the results in Figure 4 the performance of the 3D decoder.

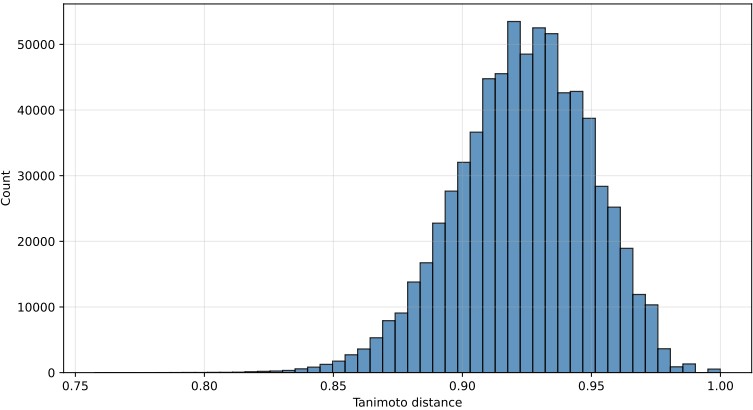

*Figure 12.* **Bioactive hit difersification experiment.** Tanimoto distances of Lit-PCBA ligands to our training set.

*Table 13.* **Ranking table for hit diversification (OOD) task.** We report mean ranks using a joint ranking scheme. Lower is better.

| Method | Validity ↓ | Hits (avg.) ↓ | Unique scaff. hits (avg.) ↓ | Max score ↓ | Synthesizable hits ↓ |
|---|---|---|---|---|---|
| SynLaD | **1.10** | **2.10** | **1.90** | 2.20 | **2.00** |
| SynFormer | **1.10** | 2.20 | **1.90** | 2.00 | 2.10 |
| REINVENT | 1.20 | **2.10** | 3.00 | **1.70** | 2.10 |
| ShePHERD | 4.00 | 2.60 | 2.20 | 3.10 | 2.30 |

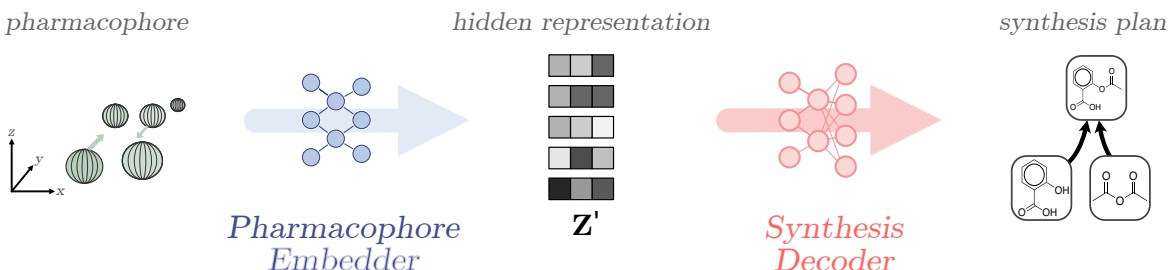

*Figure 13.* To assess the importance of our model's autoencoder structure and 3D components, we run an ablation where we remove the 3D branch of our model. Here, as shown, we instead train a model to map directly from a pharmacophore to a synthesis plan via a deterministic hidden representation, $\mathbf{Z}'$. For consistency with SynLaD, we keep pharmacophore embedding network the same as that used as part of the conditional diffusion transformer, and we also ensure that the hidden representation $\mathbf{Z}'$ has the same dimensionality as the previously used stochastic latent variable, $\mathbf{Z}$. The synthesis decoder also decodes from $\mathbf{Z}'$ in the same way as before, using cross-attention (see Figure 1, C).

## C.3. Ablations

**Removing the 3D VAE and diffusion module.** To see whether there is benefit in modeling the conditioning input to the synthesis decoder using a latent diffusion framework, we perform an ablation where we remove the 3D latent component from SynLaD and condition the synthesis decoder directly on the pharmacophore. We preserve the pharmacophore embedder described in Section 3.2 and use the resulting embedding to condition on with an otherwise identical synthesis decoder via cross-attention. This yields a simplified pipeline, pharmacophore → pharmacophore embedding → synthesis pathway (see Figure 13), which isolates the contribution of the 3D branch while keeping the synthesis decoder architecture and training procedure unchanged. We run the ablation on the in-distribution (ID) and out-of-distribution (OOD) conditioning tasks reported in Section 4 and show in Table 14 that although the Tanimoto Combo scores do not suffer a drastic drop, the ablated version of the model appears to have collapsed to generating few samples per input (see Figure 14), showing that the entire framework is crucial to generating diverse hits.

**Using a SMILES VAE.** We further investigate the effects of learning a conditional diffusion model for phramacophore-conditioned generation in a less expressive latent space. We replace the 3D autoencoder with a transformer VAE that operates on SMILES strings, to remove the 3D-reasoning component of SynLaD, while keeping the latent-conditioning component. SMILES strings are tokenized at the atom level using a regular-expression tokenizer adapted from Schwaller et al. (2018). The SMILES VAE preserves the encoder-decoder architecture of the 3D autoencoder, and the same per-token diagonal-Gaussian latent. $\mathcal{L}_{3D}$ in Eq. (1) is replaced by a cross-entropy loss over tokens, and $\lambda$ is set to 0.5. We keep an identical synthesis decoder, and train a second-stage latent diffusion model conditioned on pharmacophore embeddings. We run the ablation on both the ID and OOD conditioning tasks in Section 4 and report results in Table 14. We note that the diversity and uniqueness advantages of SynLaD are preserved, supporting our hypothesis that learning a pharmacophore-conditional distribution over latents via the diffusion module helps, promoting variability in the synthesis decoder's conditioning. In contrast, the number of hits and Tanimoto scores decrease, suggesting that 3D information helps better capture the latent space for the pharmacophore-conditioning task.

## C.4. Comparison to baselines

**ShEPhERD** We used the official implementation at `https://github.com/coleygroup/shepherd` (Adams et al., 2025) and use the $p(x_1|x_3, x_4)$ conditional setting provided in the repository where $x_1$ denotes molecular structure, $x_3$

*Table 14.* **Ablations.** We report metrics for our in-distribution (ID) and hit diversification / out-of-distribution (OOD) experiments. *SynLaD (syn out)* refers to synthesis output of the baseline SynLaD model, *w/o 3D VAE & DM* refers to an ablation where we remove the 3D heads and diffusion module and condition the synthesis decoder directly on pharmacophore embeddings, and *w/ SMILES VAE* refers to a model where we switch the 3D VAE for a SMILES VAE. Metrics are **averages** over all conditioning inputs.

| Setting | Method | Div. ↑ | Uniq. ↑ | Tanimoto Combo ↑ | #Hits ↑ | #Scaff. hits ↑ |
|---------|--------|--------|---------|------------------|---------|----------------|
| ID | SynLaD (syn out) | 0.75 | 0.63 | 1.18 | 27.5 | 11.6 |
| ID | w/o 3D VAE & DM | 0.49 | 0.17 | 1.12 | 5.9 | 3.5 |
| ID | w/ SMILES VAE | 0.72 | 0.55 | 1.07 | 20.7 | 8.9 |
| OOD | SynLaD (syn out) | 0.86 | 0.79 | 1.29 | 17.9 | 6.9 |
| OOD | w/o 3D VAE & DM | 0.76 | 0.18 | 1.11 | 4.4 | 1.5 |
| ID | w/ SMILES VAE | 0.86 | 0.77 | 1.19 | 10.4 | 3.2 |

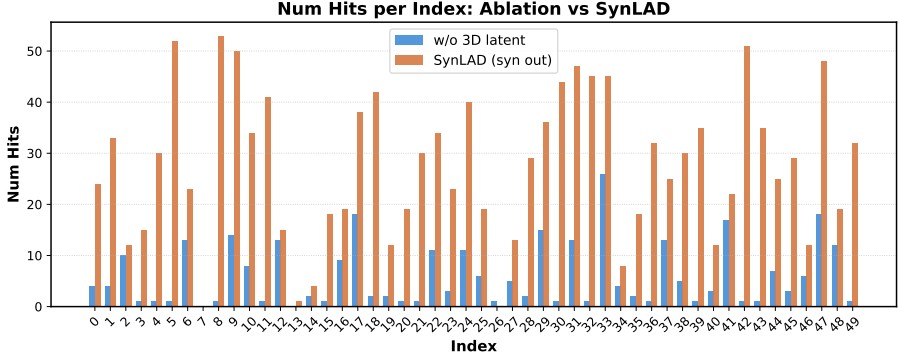

*Figure 14.* **Effect of removing the 3D branch on the number of hits.** We generate molecules conditioned on a pharmacophore with 1) SynLaD and a 2) synthesis decoder directly conditioned on the pharmacophore embedding.

is the query electrostatic potential surface, and $x_4$ is the query's pharmacophore. We generate 500 analogues with number of atoms uniformly sampled from the interval $[\max(3, N - 6), N + 6]$, where $N$ is the number of atoms in the query molecule (including hydrogens), to best match SynLaD's settings.

**SynFormer** We used the official implementation at `https://github.com/wenhao-gao/synformer` and changed the following inference settings to allow for higher quality designs compared to the default: `search_width=32`, `exhaustiveness=128`, `time_limit=300`. We generate and evaluate 500 molecules per query.

**REINVENT** We used the official implementation of REINVENT 4 available at `https://github.com/MolecularAI/REINVENT4` (Loeffler et al., 2024; Blaschke et al., 2020; Olivecrona et al., 2017). We start from the default REINVENT prior (from `https://zenodo.org/records/15641297`), which was trained on ChEMBL (Mendez et al., 2019) to cover a broad region of chemical space. We followed the repository's example reinforcement learning configuration (including hyperparameters), using a Tanimoto color scorer (with up to 5 conformers for computational speed) and a maximum of 300 steps (we do not include any other scoring components). We use the final 500 sampled molecules for evaluation. Although additional steps and hyperparameter tuning may further improve results, each target run already takes approximately 6 hours and can involve scoring up to 30 000 proposed molecules; therefore, we defer a comprehensive hyperparameter study (for both REINVENT and SynLaD) to future work.

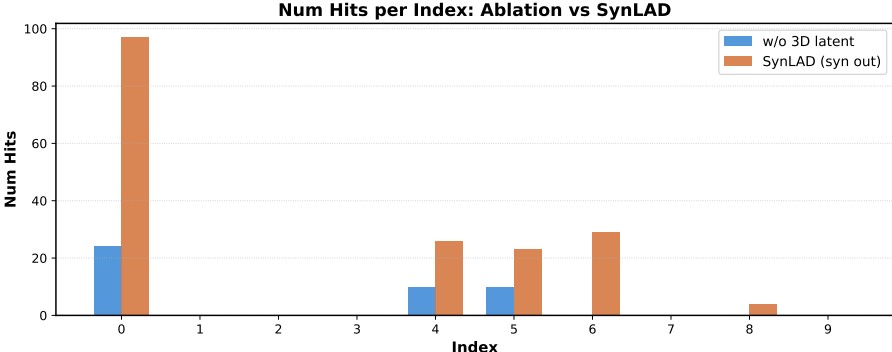

*Figure 15.* **Effect of removing the 3D branch on the number of hits, for our OOD hit diversification experiment.**
We replicate the setting in Figure 14.

