# OpenReview forum: "SynLaD: Latent Diffusion for Generating Synthesizable Molecules Conditioned on 3D Pharmacophore Profiles"
_ICML.cc/2026/Conference — ICML 2026 regular_

### Official Review · Reviewer_YhQC · 2026-02-17

**Soundness:** 3
**Presentation:** 3
**Significance:** 3
**Originality:** 3
**Overall Recommendation:** 5
**Confidence:** 4

**Summary:**

The paper presents SynLaD that learns a shared latent representation that can be decoded into both a molecular 3D structure and a plausible synthesis pathway. Novel molecules are generated by applying diffusion on this shared latent space, conditioned on the pharmacophore of a query molecule. The model is evaluated on analog generation tasks.

**Compliance With Llm Reviewing Policy:**

Affirmed.

**Final Justification:**

Thanks the authors for the update. As my concerns were fully resolved, I changed my score accordingly and recommend the paper for acceptance.

**Key Questions For Authors:**

### Weaknesses/Questions:
1. The claim that the 3D component is crucial for the performance of SynLaD is not well supported.
    1. Could the authors describe the ablation from Appendix B.3 in more detail? What is the "3D latent component"? I assume it is the entire $Z$, as $Z$ is homogeneous ($Z \in R^{d \times N}$). If $Z$ is removed, how is the diffusion performed?
    2. Removing the molecule decoding part is a drastic ablation and does not decouple only the 3D information, but the structure reconstruction as well. Could the authors provide an ablation using a 2D auto-encoder or the same 3D auto-encoder trained on flattened molecules (representing 3D molecules in 2D)? This would disable 3D reasoning while leaving structure reconstruction intact.
    3. Could the authors evaluate the ablated model in an out-of-distribution (OOD) setting?
2. There is no limitations section; could the authors provide one?
3. The cross-decoder agreement is quite low, likely due to the poor reconstruction capabilities of the synthesis head (Table 1). Do the authors have ideas on how to improve this reconstruction? Is this a limiting factor of SynLaD?
4. OOD query molecules are not compared to training data. Could the authors provide a histogram of Tanimoto similarity between query molecules and the training set to confirm the setting is indeed OOD?
5. The results in Table 4 are dominated by a single query molecule (as noted by the authors). Could the authors provide the mean ranks for all metrics instead of the mean of absolute values? This would better show how often SynLaD outperforms other methods.
6. It is not clear how the number of atoms is selected during inference.

**Limitations:**

There's no limitation section

**Strengths And Weaknesses:**

### Strengths:
* Analog generation is an important and relevant topic.
* Decoding both a synthesis path and a 3D structure from a shared latent is an elegant idea, as is conditioning the diffusion on pharmacophores.
* The paper is well-written and structured. I especially liked the Experiments section, where each setting/characteristic of the model has its own subsection.
* The results look promising, with both decoding heads appearing to support each other.

### Main Weakness:
The primary weakness of the paper is that the claim that the 3D component is crucial for the performance of SynLaD is not well supported. While this is the core of the paper’s narrative, I am lowering my score until the authors clarify this claim or provide more supporting experiments. Detailed questions regarding this weakness and other minor points are in the Questions section. Once these are addressed, I’ll be happy to raise my score to 5.

---

> ### Author Rebuttal · Authors · 2026-03-30
>
> We thank the reviewer for their thoughtful feedback and address each point below, clarifying our design choices regarding the 3D component.
>
> ### Q 1.1
> **Further details on how ablation is run** In Appendix B.3, we remove both the 3D encoder/decoder and the diffusion module, and condition the synthesis decoder directly on pharmacophore embeddings. Explicitly, if $S=(s_1,\dots,s_L)$ denotes the synthesis sequence and $c$ the conditioning pharmacophore, the SynLaD model is:
>
> $p_{\mathrm{SynLaD}}(S \mid c) = \int p_{\theta}(S \mid Z) p_{\phi}(Z \mid c)  dZ$,
>
> where the distribution over $Z$ is learned by the pharmacophore-conditioned latent diffusion model. In contrast, in Appendix B.3, we replace the stochastic latent $Z$ with a deterministic pharmacophore embedding $\tau(c)$, therefore:
>
> $p_{\mathrm{abl}}(S \mid c)=p_{\mathrm{abl}}(S \mid \tau(c)) $.
>
> This is a direct pharmacophore-to-synthesis model where the decoder cross-attends to a deterministic $\tau(c)$ instead of a sampled $Z$. This isolates the contribution of the learned 3D latent while keeping the synthesis decoder and training unchanged. We will add a schematic to clarify this.
>
> **Why ablation was run and intuition behind results** Analogue generation is one-to-many: a single pharmacophore can map to many valid molecules and synthesis pathways. To evaluate our modeling choice, we compare against a model that directly maps pharmacophore to synthesis.
>
> With direct conditioning, the decoder receives a deterministic $\tau(c)$, leading to a narrower mapping and reduced diversity. While the average Tanimoto Combo drops modestly ($1.18 \rightarrow 1.12$), diversity, uniqueness, hit count, and scaffold hits degrade significantly ($0.75 \rightarrow 0.49$, $0.63 \rightarrow 0.17$, $27.5 \rightarrow 5.9$, $11.6 \rightarrow 3.5$; see Appendix B.3 and Figure 12).
>
> We therefore find that our learned latent 3D space enables a multimodal, pharmacophore-conditioned distribution that supports diverse analogue generation.
>
> ### Q 1.2
> Our use of a 3D autoencoder is motivated by the conditioning signal (pharmacophores, defined by types and 3D positions) being inherently 3D, making point clouds the information-complete representation. A 2D autoencoder would remove precisely the structural information we aim to model and not be a faithful proxy.
>
> While such an ablation could be informative, it would require a substantially different architecture (currently we use a non-equivariant transformer reconstructing atom types and 3D coordinates) and a full training and tuning cycle, which is unfortunately not feasible within the rebuttal timeline.
> ### Q 1.3
> We now include results for the ablated model in the OOD setting in a new table and figure, summarizing the results below:
>
> | Model | Hits (avg.) |
> |-|-|
> | SynLaD | 17.9 |
> | w/o 3D latent | 4.4 |
>
> We note that the findings from the in-distribution experiments are preserved.
> ### Q 2
> We are happy to add an explicit limitations section, consolidating limitations and future directions discussed throughout the paper. This includes exploring alternative or multiple reaction predictors for robustness, scaling to larger datasets, and evaluating more challenging OOD splits. We will also incorporate points raised in the review, such as reliance on external reaction predictors and improving cross-decoder agreement.
> ### Q 3
> Several directions could improve cross-decoder agreement: (i) adding a cross-decoder consistency loss (e.g., penalizing mismatches between decoded products), (ii) improving or ensembling reaction predictors, (iii) pretraining the synthesis head to strengthen reconstruction. In our answers to R1&2 we also outline why we believe only semantic agreement, as opposed to full agreement is more important.
> ### Q 4
> This is a great suggestion. We have computed and will include a histogram of Tanimoto distances between Lit-PCBA ligands and training molecules (using 2048-bit Morgan fingerprints) in the appendix. Aggregated across queries, the minimum distance is 0.75 and the median is ~0.92. Unfortunately, we cannot upload the figure in this response.
> ### Q 5
> Below we report the rankings (only for now on the number of synthesizable hits due to the limited rebuttal space), computed using a joint ranking scheme.
>
> | Method | Mean Rank ↓ |
> |-|-|
> | SynLaD | **2.00** |
> | SynFormer | 2.10 |
> | REINVENT | 2.10 |
> | ShePHERD | 2.30 |
>
> ### Q 6
> For the unconditional setting, the number of atoms is sampled from the empirical distribution of molecule sizes observed in the training data. For the conditional setting, where we condition on the pharmacophore of a query molecule, we sample the number of atoms for the generated latent from $[max(1, N − 3), N + 3]$, where $N$ is the number of atoms in the query -- see Appendix B.1 (we'll provide a more explicit link to this in the main text).
>
> [1] Gao, W. et al., Generative AI for navigating synthesizable chemical space, PNAS 2025

---

> > ### Author Rebuttal · Reviewer_YhQC · 2026-04-03
> >
> > While all the minor concerns have been addressed, a key point remains:
> > - Ablation via Flattened 3D Inputs: I acknowledge that training a 2D auto-encoder from scratch is not feasible during the rebuttal period. However, I previously suggested running ablations on flattened 3D molecules using the existing 3D auto-encoder. This approach requires no architectural changes or retraining but would effectively isolate the impact of 3D reasoning while keeping the structural reconstruction intact. Without that (or some similar less drastic ablation), it’s hard to access the main claim of the paper that the 3D component is crucial for the performance of SynLaD.
> >
> > Moreover, could the authors provide the full ranking table? I believe this data would offer a more comprehensive view of the model's performance and can be easily included in the discussion.

---

> > > ### Author Response · Authors · 2026-04-07
> > >
> > > We thank the reviewer for their feedback and are glad to hear we addressed several of their initial concerns. Below we discuss the remaining two concerns.
> > >
> > > ### 2D Autoencoder Ablation
> > >
> > > Thanks again for the suggestion! We performed the ablation by retraining the VAE to reconstruct flattened molecules (an *inference-only* ablation was not possible because SynLaD does not take a molecular structure as input that could be flattened; rather the diffusion model directly generates a latent in the learned space of the frozen VAE; therefore flattening coordinates was only possible during training). Molecules were flattened by zero-ing the z-coordinate and, following the scheme described in Section 3.1, we then trained the diffusion module using 2D-derived latents. We add results (last row) to the existing metrics in Appendix Table 11 (for ID task:):
> > >
> > > | Method | Div. ↑ | Uniq. ↑ | Tanimoto Combo ↑ | #Hits ↑ | #Scaff. hits ↑ |
> > > |------------------|--------|---------|------------------|---------|----------------|
> > > | SynLaD (syn out) | 0.75 | 0.63 | 1.18 | 27.5 | 11.6 |
> > > | w/o 3D VAE & DM | 0.49 | 0.17 | 1.12 | 5.9 | 3.5 |
> > > | w/ 2D VAE | 0.75 | 0.68 | 1.07 | 20.9 | 8.6 |
> > >
> > > And for OOD task:
> > >
> > > | Method | Div. ↑ | Uniq. ↑ | Tanimoto Combo ↑ | #Hits ↑ | #Scaff. hits ↑ |
> > > |------------------|--------|---------|------------------|---------|----------------|
> > > | SynLaD (syn out) | 0.86 | 0.70 | 1.30 | 17.9 | 6.9 |
> > > | w/o 3D VAE & DM | 0.64 | 0.19 | 1.11 | 4.4 | 1.5 |
> > > | w/ 2D VAE | 0.86 | 0.87 | 1.13 | 12.9 | 4.4 |
> > >
> > >
> > > We note that the diversity and uniqueness advantages of SynLaD are preserved, supporting our hypothesis that learning  $p_{\phi}(Z | c)$ via the diffusion module helps, promoting variability in the synthesis decoder’s conditioning. In contrast, the number of hits and Tanimoto scores decrease, suggesting that 3D information helps better capture the latent space for the pharmacophore-conditioning task. As a caveat, we note that a more principled 2D ablation—using a VAE tailored to 2D graph reconstruction, with more principled tuning—will be included in the final manuscript.
> > >
> > > Finally, we would like to clarify a point regarding our use of the term “3D.” Our claims that removing the 3D head harms performance are not solely about geometric reasoning (indeed, the 3D head, as currently referred to in the manuscript, contains the reconstruction task **and** the diffusion module), but about the role of a **learned structural latent** obtained *via reconstruction*. In SynLaD, we hypothesize that this component provides a richer conditioning signal for the synthesis decoder than a raw pharmacophore, and indeed we find it is critical for achieving both diversity and quality in generated analogues (via the ablation in Appendix B.3).
> > >
> > > We therefore agree that referring to this component purely as “3D” is imprecise, and a more accurate description is a **structure-aware latent component** learned through reconstruction. We thank the reviewer for pointing out this ambiguity and will revise the wording accordingly.
> > >
> > > ---
> > > ### Ranking table
> > > As the character count has reset for the discussion, we are happy to provide the full ranking table below (and will add to the paper’s appendix):
> > >
> > >
> > > | Method    | Validity ↓ | Hits (avg.) ↓ | Unique scaff. hits (avg.) ↓ | Max score ↓ | AiZynth ↓ | Synthesizable hits ↓ |
> > > |-----------|------------|----------------|------------------------------|-------------|-----------|----------------------|
> > > | SynLaD    | **1.10**   | **2.10**       | **1.90**                     | 2.20        | **1.0**  | **2.00**             |
> > > | SynFormer | **1.10**   | 2.20           | **1.90**                     | 2.00        | 3.00      | 2.10                 |
> > > | REINVENT  | 1.20       | **2.10**       | 3.00                         | **1.70**    | **1.0**      | 2.10                 |
> > > | ShePHERD  | 4.00       | 2.60           | 2.20                         | 3.10        | 4.00      | 2.30                 |
> > >
> > > AiZynth is evaluated globally, while other metrics are averaged across pharmacophores.

---

### Official Review · Reviewer_AHbo · 2026-03-04

**Soundness:** 3
**Presentation:** 4
**Significance:** 3
**Originality:** 3
**Overall Recommendation:** 5
**Confidence:** 4

**Summary:**

The work addresses the problem of generating pharmacologically relevant molecules that are synthesizable. It does so by using an autoencoder framework, with a generator mapping 3d molecular structures to a latent space that can be decoded to both 3d molecular structures and synthesis pathways simultaneously. A diffusion transformer is used to perform generation in this latent space learned by the autoencoder.

**Compliance With Llm Reviewing Policy:**

Affirmed.

**Final Justification:**

In my opinion, the rebuttal addressed the main concerns I had, and I didn't see any point raised by other reviewers that I found critical. As stated in my rebuttal acknowledgement, I believe that the benefits of the paper outweigh the shortcomings of the paper, especially since the domain of generating synthesis paths is very challenging, so there is no easy way to avoid the shortcomings; they are somewhat inherent to the difficulty of domain itself. Therefore, I maintain my positive rating for acceptance.

**Key Questions For Authors:**

1. Is there a reason why you use the synthesis path only as output, but not as an input? An auto-encoder on both modalities simultaneously seems like an obvious choice, so it's strange to me that this was not addressed. Answering this sufficiently would affirm the modeling choice made in the paper [Soundness].
2. How important is the quality of the synthesis decoder versus the reaction prediction oracle? This could clarify the errors between synthesis path and 3D prediction [Soundness]

**Limitations:**

I encourage the authors to expand the Impact Statement. Given the application to pharmacophores, which are *intended for use in humans*, I feel that it's appropriate to be a bit more specific about the positive and negative consequences this has.

**Strengths And Weaknesses:**

# Strengths
- The problem of generating synthesizable molecules is highly relevant to the field to make ML techniques for chemistry more applicable. [Significance+]
- To my knowledge, this is a novel combination of techniques and models. It is a timely contribution with the rise of latent diffusion models in other domains. The autoencoder setup to define a good latent space is also conceptually simple, meaning that the ideas from it may be usable in other domains as well. [Originality+]
- Appropriate experimental ablations are performed to show that all the components in this setup is beneficial, such as the jointly trained prediction heads [Soundness+]
- Experiments are thorough and evaluate meaningful aspects of the proposed system. I like the fact that both downstream performance, as well as representation quality itself are evaluated. There are also quite a number of interesting evaluations in the appendix, such as the performance of the reaction predictor oracle. [Soundness+]
- The writing is clear and I had no trouble following the narrative [Presentation+]

# Weaknesses
- I don't think there is a metric that evaluates if the predicted reaction path itself is "correct" (assuming it is unique), only metrics on the end result of the synthesis path. As such, it's difficult to tell whether the consistency between 3D and synthesis decoders can fail due to mistakes by the synthesis decoder itself, or by mistakes of the reaction prediction oracle. [Soundness-]
- Given the critical reliance on the reaction prediction oracle for evaluating the synthesis decoder, I would have liked to see a more in-depth analysis of it. I see that Table 9 shows the accuracy of the reaction predictor itself, but given it is used multiple times for a synthesis path, errors may compound and the prediction is essentially on OoD data (since it is generated by the synthesis decoder). [Soundness-]
- DAGs only define a partial order, not a total order, so there is no *unique* serialization of the synthesis path. It was not clear to me if/how this problem is dealt with, and what effect this would have on the modeling of it. Another factor that makes a DAG non-unique is that there may be several viable paths towards a product, only one of which is presumably in the dataset. [Presentation-]

---

> ### Author Rebuttal · Authors · 2026-03-30
>
> We thank the reviewer for their thoughtful comments and suggestions, and address each of the concerns below. We remain open to any questions and further clarifications.
> ### Weakness 1 (Evaluation of synthesis correctness)
> We agree that ideally one would evaluate the correctness of synthesis pathways directly. However, this is challenging due to non-uniqueness: multiple valid routes can exist for the same molecule, making exact sequence matching unreliable.
>
> Following prior work, we evaluate synthesizability via the end product using AiZynthFinder, which provides an independent retrosynthetic validation signal. Nonetheless, we agree with the reviewer that reporting additional synthesis sequence generation metrics could be valuable, and we thank them for this suggestion. We now additionally report sequence-level metrics on the test set: token accuracy (overall per-token correctness), forward selection (reaction token accuracy), and building block (BB) selection (reactant accuracy). These metrics are invariant to ordering differences (i.e., permuted but equivalent plans are not penalized), but do not account for alternative valid routes.
> | Metric | Value |
> |-|-|
> | Token accuracy | 0.83 |
> | Forward selection | 0.99 |
> | BB selection | 0.77 |
>
> These indicate strong performance on key components. While errors from the decoder or oracle exist, strong AiZynthFinder results suggest they do not significantly impact practical validity.
> ### Weakness 2 (Reaction predictor analysis)
> We agree that exploring where these reaction predictors work and don’t work (particularly on OOD data) is an interesting avenue to explore. Unfortunately, this is somewhat out of scope of our work here and has been previously investigated elsewhere in the literature (e.g., [1,2,3,4]). In some sense we hope the AiZynthFinder metric (which uses an independent, one-step backwards predictor) captures a notion of if the errors compound and molecules we suggest are not synthesizable. However, we hope to carry out a further analysis of how reliable the reaction predictor is as a function of step length in the future.
> ### Weakness 3 (DAG serialization)
> Where invariances exist, we train on randomly generated serializations (e.g., we could switch two building blocks coming into a two reactant reaction with no meaningful effect on the final product). This happens when assembling each minibatch, so a different possible serialization is picked each time; this augmentation is similar to how the 3D invariances are dealt with for the 3D head. We do use heuristics at training time (e.g., early building blocks come first), which seem to help with downstream performance, but given more data even such refinements might be unnecessary.
>
> Yes, we only use one DAG for each possible product, and it is true that there may be alternative credible paths. We would like to explore using these in the future (we see this as a useful form of augmentation, in a similar way that it is useful to train on different paragraphs explaining the same context in natural language). This somewhat relies on having access to larger reaction datasets (e.g., Pistachio), where the greater number of reactions increases the likelihood that valid alternative paths will exist when composing them to form plans.
>
> These are interesting points and we are grateful to the reviewer for bringing this up. We will add a better discussion on this point to the appendix.
> ### Why synthesis is output-only
> We agree this would be interesting to explore (yet hard to complete during the rebuttal due to time).
>
> To give more context on our modeling choice: we treat the 3D component as an **inductive bias** shaping the latent space, while the synthesis pathway acts as a **hard constraint** ensuring generated molecules are synthesizable. Concretely, the 3D encoder/decoder learns a latent capturing spatial and pharmacophore-relevant structure, which guides generation toward desired 3D properties. At inference, we rely on the synthesis decoder, as it provides explicit synthesizability guarantees.
>
> Including synthesis as an input modality would impose a much stronger constraint on the latent space, whereas our goal is to learn a flexible 3D-conditioned chemical representation. This asymmetry is intentional: the model learns a **3D-informed latent prior over synthesis trajectories**, rather than symmetrically encoding both modalities. We will add further clarification in the paper.
>
> [1] Kovács, D. P. et al., Quantitative interpretation explains machine learning models for chemical reaction prediction and uncovers bias. Nat. Commun. 12, 1695 (2021).
>
> [2] Gil, V. S. et al. Holistic chemical evaluation reveals pitfalls in reaction prediction models. arXiv 2312.09004 (2023).
>
> [3] Bradshaw, J. et al. Challenging Reaction Prediction Models to Generalize to Novel Chemistry. ACS Cent. Sci. 11, 539–549 (2025).
>
> [4] Tanovic et al. An exploration of dataset bias in single-step retrosynthesis prediction. ChemRxiv. 30 July 2025.

---

> > ### Author Rebuttal · Reviewer_AHbo · 2026-03-31
> >
> > I have read all the reviews and rebuttals. While there are some unresolved limitations regarding the reliance on the reaction predictor, my perspective is that this is inherent to the difficulty of the task and just the complexity of reality that has to be dealt with.
> >
> > I believe this to be a relatively minor issue and not detracting from the main contributions of the paper. Therefore, I maintain my positive rating for acceptance.

---

> > > ### Author Response · Authors · 2026-04-06
> > >
> > > We thank the reviewer for their feedback and for maintaining a positive assessment. In the revised manuscript, we will incorporate clarifications and expand the discussion accordingly.

---

### Official Review · Reviewer_gjLq · 2026-03-06

**Soundness:** 3
**Presentation:** 3
**Significance:** 2
**Originality:** 2
**Overall Recommendation:** 3
**Confidence:** 5

**Summary:**

The paper proposes SynLaD, a generative framework for unifying 3D structure-based molecular design with synthetic accessibility. It uses a two-headed variational autoencoder to learn a shared continuous latent space, from which separate decoders reconstruct 3D molecular structures and serialized synthesis pathways. A pharmacophore-conditioned flow matching model, parameterized by a Diffusion Transformer (DiT), is then trained in this latent space to generate novel molecules from 3D pharmacophore embeddings. Empirically, the paper shows that joint training improves the synthesizability of 3D-decoded molecules and reports strong, practically relevant results on pharmacophore-conditioned generation and bioactive hit diversification relative to existing baselines.

**Compliance With Llm Reviewing Policy:**

Affirmed.

**Final Justification:**

My final recommendation remains weak reject. While the paper addresses a relevant problem and reports encouraging empirical results, I do not think the current version supports its central methodological claim strongly enough. In particular, the results show that the 3D-informed latent helps synthesis generation, but they do not yet convincingly demonstrate strong shared-latent alignment between the 3D and synthesis branches: the reported agreement is still limited, and the appendix examples show clear mismatches. This remains my main concern and is why I keep significance and originality only moderate.

The rebuttal was useful in clarifying that the intended goal is semantic rather than exact alignment, and the follow-up discussion of the reaction-prediction component is informative. However, these clarifications do not materially change my assessment. The evidence still suggests only partial coupling between the two modalities, and the contribution of the external reaction predictor to final synthesis performance is still not fully disentangled in an end-to-end sense. Therefore, I keep my original recommendation unchanged.

**Key Questions For Authors:**

Please see the weaknesses above; my main questions are about (i) the degree of methodological novelty beyond integrating existing components, (ii) the extent to which the shared latent actually aligns the 3D and synthesis branches, and (iii) the extent to which the final synthesis results rely on the external reaction oracle at inference time.

**Limitations:**

No. The Impact Statement is too brief. The authors should more explicitly discuss dual-use risks and the practical limitations of the claimed synthesizability.

**Strengths And Weaknesses:**

**Strengths**：
1. **The paper addresses an important and well-motivated problem setting.** Jointly modeling *what to make* (3D pharmacophore-aligned molecules) and *how to make it* (reaction-based synthesis pathways) is a practically meaningful objective for molecular design, and the paper frames this motivation clearly.

2. **The empirical results provide credible evidence that the 3D-informed latent is useful for synthesis generation.** The paper does more than propose a combined architecture: its ablation shows that removing the 3D latent substantially hurts diversity, uniqueness, hit count, and scaffold hit count, suggesting that the 3D component contributes meaningfully to conditional synthesis generation rather than serving as a cosmetic auxiliary branch.

3. **The method shows promising practical value in downstream conditional-generation settings.** In the screening case study, SynLaD achieves higher hit counts than brute-force ROCS screening while using roughly two orders of magnitude fewer samples. More broadly, the paper reports a strong overall balance in the bioactive hit diversification benchmark, with high numbers of synthesizable hits and good query coverage.

**Weaknesses**：

1. **The contribution appears more integrative than fundamentally novel.** The method combines pharmacophore-conditioned latent generation, 3D decoding, and synthesis-pathway decoding in a shared latent space, but each of these components is already motivated by prior work. As a result, the paper reads more as a strong unification of existing ideas than as a clearly new modeling principle.

2. **The shared latent does not yet enforce strong cross-decoder consistency.** A key claim is that the same latent should support both 3D and synthesis decoding for the same molecule, yet the reported agreement remains limited and the appendix shows clear mismatches between the two outputs. The 3D branch is also noticeably less stable than the synthesis branch under conditioning, suggesting only partial coupling between the two modalities.

3. **The synthesis pipeline is not fully end-to-end at inference time.** Product identities are not generated directly by the synthesis decoder, but inferred by a separately trained reaction-prediction oracle. This introduces an additional error source into multi-step generation, and the paper does not clearly isolate how much of the final synthesis quality or 3D/synthesis consistency depends on this external component.

---

> ### Author Rebuttal · Authors · 2026-03-30
>
> We thank the reviewer for their feedback and address each concern below.
> ### Weakness 1 (Novelty)
> While our work builds on prior components, **the core contribution is not merely integrative**. We introduce a **joint latent space constrained simultaneously by 3D structure and synthesis pathways**, trained jointly. To our knowledge, learning a shared latent that must decode into both a valid 3D conformation and an executable synthesis route has not been explored before. This requires **non-trivial optimization trade-offs** (e.g., balancing geometric fidelity and sequence likelihood; Appendices A.2–A.4).
>
> Additionally, we introduce **pharmacophore-conditioned latent generation in a non-equivariant architecture**, which is also unexplored. Prior work either relies on equivariant models or does not support conditioning on rich spatial features in latent space (Appendices A.6, B.1).
>
> Finally, our experimental results demonstrate that the approach yields non-trivial emergent benefits: the jointly trained model outperforms both synthesis-only and 3D-only baselines across in- and out-of-distribution settings, improving the trade-off between synthesizability, diversity, and pharmacophore alignment. These gains are not expected from a naïve combination of components, but instead arise from the learned coupling between modalities in the latent space.
> ### Weakness 2 (Decoder agreement)
> We acknowledge that the two decoders do not always agree on the exact chemical identity of their outputs. However, we would like to clarify that exact matching **is not an objective of our model**. Instead, our goal is to achieve **semantic alignment** between the two modalities, such that they agree on key structural and functional properties relevant to the task. We will clarify this in the manuscript.
>
> In practice, this is reflected through partial agreement metrics—such as scaffold overlap, shape similarity, and pharmacophore consistency (Figure 2), as well as qualitative examples (Appendix Figures 7 and 9)—which demonstrate that both decoders capture shared, task-relevant features. This level of alignment is sufficient for our purpose: learning a latent space that conditions the synthesis decoder on 3D information for pharmacophore-guided generation.
>
> Importantly, enforcing exact correspondence between two heterogeneous decoders—one operating in continuous 3D space and the other in discrete synthesis space—is a fundamentally challenging problem. More broadly, ensuring that generated molecules map to synthesizable space is itself a well-known difficulty in molecular generative modeling [1].
>
> In this context, we emphasize that our model achieves **meaningful partial synchronization**, which is already non-trivial. As shown in Figure 2 and Table 2, joint training improves the synthesizability of molecules produced by the 3D decoder, indicating that the shared latent space successfully couples geometric and synthetic information.
>
> Crucially, this partial alignment translates into improved downstream performance: in pharmacophore-conditioned generation—our primary use case—the model produces more synthesizable, diverse, and pharmacophore-consistent hits, as demonstrated in our experimental results.
> ### Weakness 3 (Reaction predictor)
> We believe that decoupling the reaction prediction task from synthesis token generation is actually an advantage: this does not allow the method to hack the reaction predictor and mix up search and single step prediction. To isolate how much of the synthesis quality/consistency results depend on the reaction prediction step, we already provided an analysis of the reaction predictor performance in Appendix Section A.7 (although we note that true accuracy is hard to measure for reaction predictors due to the fact that for some reactants there may be multiple possible correct products but only one is recorded in the ground truth data).
>
> While the reaction predictor does introduce an additional source of error, we emphasize that the objective of our method is not to perfectly model reaction outcomes, but rather to improve overall synthesizability of molecule hits relative to strong baselines (e.g., as judged by AiZynthFinder which uses a separate, template-based backwards reaction predictor to assess synthesizability). In this regard, our results demonstrate that, despite this potential source of noise, the proposed approach consistently achieves our objective.
>
> ### Impact statement
> We thank the reviewer for pointing this out and we are including a more comprehensive Impact Statement, highlighting the potential dual-use risks of our method as well (e.g., around possible misuse in proposing harmful or toxic compounds).
>
> **Finally, we remain open to any further questions and feedback that may improve our work!**
>
> [1] Gao, W. et al.,”The Synthesizability of Molecules Proposed by Generative Models”, J. Chem. Inf. 2020

---

> > ### Author Rebuttal · Reviewer_gjLq · 2026-04-03
> >
> > Thank you for the rebuttal. I appreciate the clarifications, and I agree that the response helps better frame the intended claims of the paper, especially regarding semantic rather than exact cross-decoder alignment. However, my concerns are only partially addressed. I still view the contribution as more integrative than fundamentally novel; even under the clarified goal of semantic alignment, the current evidence suggests only partial coupling between the 3D and synthesis branches relative to how central the shared-latent claim is; and the paper still does not clearly isolate how much of the final synthesis results depend on the external reaction predictor at inference time. Because these remaining concerns go to the core methodological and empirical claims of the paper and would require more substantial revision rather than a short rebuttal to address, I selected option (c) and keep my original score unchanged.

---

> > > ### Author Response · Authors · 2026-04-07
> > >
> > > We thank the reviewer for their continued engagement. Firstly, **regarding novelty**, we underscore that, as far as we are aware, SynLaD is the first method to learn coupled, aligned latent spaces of 3D molecules and their synthesis pathways, using an autoencoder to tie these important modalities together. Furthermore, we demonstrate how we can learn a conditional diffusion model in SynLaD’s latent space, showing that such a framework leads to SOTA pharmacophore-conditioned analogue generation.
> > >
> > >
> > > Below we address the two other remaining concerns (and are pleased that the others are resolved):
> > >
> > > > even under the clarified goal of semantic alignment, the current evidence suggests only partial coupling between the 3D and synthesis branches relative to how central the shared-latent claim is
> > >
> > > We understand that our original wording may have caused confusion, and we will revise the manuscript to clarify our claim regarding the shared-latent, and instead explain that the goal of the autoencoder is to achieve **semantic alignment, which suffices in our task** of enforcing agreement on key structural (including in 3D) and functional features for pharmacophore-conditioned analogue generation. Our manuscript presents strong evidence that this goal is achieved (Fig. 2 and *Consistency Analysis Section*, Appendix Figs. 7 and 9), and that this leads to high quality, diverse generated molecules in our main task, achieving very competitive results to both baselines (Section 4.3, Table 4, Figure 4) and brute-force library screening (Section 4.3, *Screening case study*, Table 3).
> > >
> > > > the paper still does not clearly isolate how much of the final synthesis results depend on the external reaction predictor at inference time
> > >
> > > We thank the reviewer for raising this point. We agree that it is important to disentangle the contribution of the external reaction predictor at inference time. We provide a breakdown of the sources of error in the synthesis pipeline, which together account for the final synthesis performance (with a reconstruction accuracy of 63.4% as reported in Table 1):
> > >
> > > - Decoder-level performance (our response to Reviewer AHbo):
> > >
> > > | Metric            | Value |
> > > |-------------------|-------|
> > > | Token accuracy    | 83.1%  |
> > > | Forward selection | 99.0%  |
> > > | BB selection      | 77.3%  |
> > >
> > > - Reaction predictor accuracy (Appendix Table 9, top-1): 84.6%
> > >
> > > This decomposition allows us to separate errors arising from (i) the learned synthesis decoder (token and building block prediction) and (ii) the external reaction predictor. (Although we again note that it is hard to assess “true” accuracies for these tasks due to the possibility of alternative, valid answers which may not match the particular answer in the ground truth data).
> > >
> > > **More specifically, to isolate the role of the reaction predictor on the final error, we can model how the reaction prediction error rate may accumulate as multiple reactions are performed.** In particular, if we assume (a) each reaction prediction task is independent and (b) the number of reaction predictions needed at inference (correlated to the size of the synthetic plans) matches the frequencies observed in our extracted, ground-truth dataset, then we computed an expected overall error rate of **26.4%** in final products from incorrect reaction predictions.
> > >
> > > We are happy to add this analysis to our manuscript, along with a detailed breakdown of the respective performance of the different model components. We also reiterate that while any current ML-based reaction predictor we use will be imperfect, **we still believe generating molecules via chemical reactions serves as a useful inductive bias for synthesizability**, a hypothesis supported empirically through our method’s superior AiZynth scores.

---

### Official Review · Reviewer_r4AP · 2026-03-09

**Soundness:** 3
**Presentation:** 3
**Significance:** 3
**Originality:** 3
**Overall Recommendation:** 5
**Confidence:** 4

**Summary:**

This paper proposes SynLaD, a latent diffusion generative model for small-molecule drug design. The core innovation of this method lies in unifying 3D molecular structure generation and synthesis pathway planning in a shared latent space through a dual-decoder architecture. Specifically, the model first uses a variational autoencoder to encode molecules into latent representations, then reconstructs atomic coordinates and types via a 3D decoder, while simultaneously generating reaction pathways through an autoregressive synthesis decoder. In the second stage, the model trains a diffusion Transformer-based conditional generation module that can sample new latent representations according to given pharmacophore features, thereby generating molecules that match target 3D shapes and possess synthesizability. Experimental results demonstrate that this method can produce more diverse and synthesizable bioactive molecular analogues than existing methods in pharmacophore-guided molecular generation tasks.

**Compliance With Llm Reviewing Policy:**

Affirmed.

**Final Justification:**

The rebuttal fully addressed my main concerns regarding experimental fairness and technical details. My prior assessment rated soundness as 2 and presentation as 2; the clarifications on REINVENT's larger sampling budget and additional ablations improve soundness to 3. Originality and significance remain at 3, as the dual-decoder contribution is meaningful for the field. The rebuttal changed my evaluation, reinforcing that this work is technically sound and worthy of acceptance. I adjust my recommendation from 4 to 5.

**Key Questions For Authors:**

1. Section 3.1.2 mentions that a reaction prediction model is required to infer products during testing, but it is not clearly stated whether this model is pre-trained and fixed or jointly trained with SynLaD. If it is used in a fixed manner, when encountering reaction types outside the training distribution, will product prediction errors accumulate and cause the entire synthesis pathway to fail?
2. How many molecules are evaluated per step in REINVENT's 300-step optimization? If the total number of evaluated molecules or time budget is controlled to be the same, can SynLaD still maintain its advantage?
3. Can you explain why conditioning directly on pharmacophore embeddings would lead to diversity collapse?

**Limitations:**

The authors briefly mention in Section 5 the future direction of scaling to larger datasets, but the following limitations are not adequately discussed:
-The current method relies on an independent reaction prediction model to infer products; if this model makes incorrect predictions, subsequent synthesis steps will continue based on the erroneous products.
-The paper does not explore whether the learned latent space possesses chemically meaningful smooth interpolation properties, that is, whether similar latent vectors correspond to molecules with similar chemical properties.

**Strengths And Weaknesses:**

Strengths
1. The dual-decoder architecture design demonstrates certain originality. Unifying 3D structure reconstruction and synthesis pathway generation in a shared latent space, and achieving coupling of both representations through joint training, provides a novel technical route for simultaneously optimizing geometric fidelity and synthetic accessibility.
2. The problem setting aligns with practical needs in drug discovery and possesses clear application value for computational chemistry and AI-driven pharmaceutical fields.
3. The method is validated in multiple scenarios with comparisons against various baselines including ROCS virtual screening, ShEPhERD, and SynFormer, making the experiments relatively comprehensive.

Weaknesses
1. When comparing with REINVENT, the latter requires approximately 6 hours using reinforcement learning while SynLaD only needs 1 minute, but the paper does not control for consistent sampling budgets or iteration numbers. Furthermore, REINVENT only hitting 3 targets may stem from the simple ROCS color score reward design rather than inherent limitations of the method itself.
2. Section 3.2 provides vague descriptions of the pharmacophore embedding mechanism, failing to explain how discrete types and continuous coordinates are uniformly encoded or how differences in pharmacophore counts across molecules of varying sizes are handled.
3. The ablation experiments are insufficient. The performance comparison and analysis of the pure synthesis decoder after removing the 3D branch are inadequate.

---

> ### Author Rebuttal · Authors · 2026-03-30
>
> We thank the reviewer for their feedback and address each concern below. We also **respectfully seek clarification regarding the Overall Recommendation score of 1**. While we acknowledge limitations, we believe this score underestimates the contribution, as the comments **do not** indicate major methodological, novelty, or clarity issues (and individual criteria are rated “fair” or higher). We would appreciate further clarification on this assessment and how we might address it.
>
> ### Weakness 1 (REINVENT comparison)
>
> We would like to clarify that, in our experiments, **REINVENT was given a substantially larger sampling budget than SynLaD** (up to 30 000 molecules vs 500 for SynLaD during the pharmacophore directed search), effectively favoring it in terms of optimization capacity (see Appendix B.4). In addition, we carefully shaped the REINVENT reward function to closely align with the conditioning signal used in SynLaD—namely, pharmacophore similarity. This choice explicitly optimizes for the presence of the conditioning pharmacophore, in contrast to objectives such as docking scores, which only implicitly capture key pharmacophore features. To the best of our knowledge, pharmacophore similarity represents the most appropriate and competitive reward for this setting.
> ### Weakness 2 (Pharmacophore embeddings)
> We will add further details. Pharmacophore types are embedded via a learned embedding lookup, while positions are projected using a linear layer. The resulting representations are combined by summation. Section 3.2 describes pharmacophore types and Appendix B.1 explains the number of pharmacophore features sampled per query.
> ### Weakness 3 (Ablations)
> We would appreciate clarification on additional ablations requested, and note that the current set was described as comprehensive by reviewers AHbo and gjLq. Existing ablations include inference methods, training strategies, losses, latent size, sampling budget, CFG weight, and removal of the 3D branch.
> We now additionally include:
> - Beam width ablation: best performance at width 5
> | Beam width | Synthesis Match Rate (%) ↑ | Tanimoto similarity ↑ |
> |-|-|-|
> | 1 | 47.6 | 0.75 |
> | 5 | 63.4 | 0.84 |
> | 10 | 59.5 | 0.82 |
>
> - Sampling ablation: increasing N (the number of independent sequences sampled) degrades performance, highlighting sensitivity
> | N  | Top-k | Synthesis Match Rate (%) ↑ | Tanimoto similarity ↑ |
> |-|-|-|-|
> | 1 | 1 | 43.9 | 0.73 |
> | 1 | 5 | 44.1 | 0.73 |
> | 1 | 10 | 43.4 | 0.73 |
> | 10 | 1 | 1.1 | 0.20 |
>
> ### Question 1 (Reaction predictor)
>
> Thanks for this question. We answer both parts separately below:
>
> **Details of the predictor.** The full description of how this is trained is included in Appendix A.7. We can include a reference to this in the main text. In brief, it is pre-trained and fixed, on the same USPTO dataset we use for extracting synthesis plans (but only on one-step reactions).
>
> **How prediction errors are dealt with.** Yes, if the decoding goes out of distribution (or even sometimes in distribution) the prediction can fail (true accuracy is hard to measure for reaction predictors due to the fact that for some reactants there may be multiple possible correct products but only one is recorded in the ground truth data). Although a gold-standard check would be to assess these routes experimentally, this is expensive and time-consuming, so (following previous work) we instead use an in silico retrosynthesis tool, AiZynth, to assess how many of the routes are realistic (note that this uses an independent, template-based backwards reaction predictor). This effort shows that although some molecules decoded by the synthesis head may not actually be synthesizable, they represent a significant improvement over those suggested by the 3D head alone, which ignores synthesizability entirely.
>
> Future work can also look at ensembles of forward predictors to gain confidence in synthesis plans (as the reaction predictor we use is an independent part of the model, we can also take advantage of better extrapolating predictors developed elsewhere).
>
> ### Question 2 (REINVENT budget)
>
> REINVENT uses ~30,000 evaluations vs. 500 for SynLaD, highlighting the efficiency of amortized generation. Increasing SynLaD’s budget (e.g., via RL fine-tuning) would likely further improve results, but is left for future work.
>
> ### Question 3 (Diversity collapse)
>
> Appendix B.3 shows that removing the 3D head and diffusion leads to collapse. Without diffusion, the model learns a **deterministic mapping** $p_{\mathrm{abl}}(S \mid \tau(c)) $, whereas SynLaD models $p(S|c) = \int p_{\theta}(S \mid Z) p_{\phi}(Z \mid c) dZ$. (See also response to Rev.YhQC)
>
> The **latent distribution** $p_{\phi}(Z \mid c)$ injects variability, preventing collapse. This is especially important for pharmacophore conditioning, where training data encourages one-to-one mappings.
>
> We will add a discussion on this to the paper.

---

> > ### Author Rebuttal · Reviewer_r4AP · 2026-04-04
> >
> > I apologize for the error in my overall recommendation score; I intended to select "4: Weak Accept" but mistakenly chose the different one. The authors' rebuttal has adequately addressed my concerns. Regarding the REINVENT comparison, the clarification that REINVENT was given a substantially larger sampling budget (30,000 vs. 500 molecules) resolves my fairness concern. The additional details on pharmacophore embeddings and the explanation of diversity collapse through latent distribution variability are satisfactory. The new beam width and sampling ablations further strengthen the experimental validation.

---

> > > ### Author Response · Authors · 2026-04-06
> > >
> > > We thank the reviewer for their engagement and positive re-assessment. We appreciate the constructive suggestions and will incorporate them in the revised manuscript.

---

### Decision · Program_Chairs · 2026-04-30

**Decision:**

Accept (regular)

**Comment:**

This paper received a strongly positive response from reviewers for both its motivation and execution. The key contribution, namely unifying 3D molecular design and synthetic accessibility in a shared latent space with dual decoders for structure and synthesis, was seen as technically meaningful and practically relevant. Reviewers also found the empirical results on analog generation and synthesizability strong, and the overall framing of coupling “what to make” with “how to make it” was considered compelling. One reviewer remained more skeptical on the degree of novelty and requested stronger ablations, but the rebuttal appears to have addressed at least part of that concern and did not change the overall positive balance.